# Common to rare transfer learning (CORAL) enables inference and prediction for a quarter million rare Malagasy arthropods

Otso Ovaskainen ●[1,2,10] ✉, Steven Winter[3,10], Gleb Tikhonov ●[2], Nerea Abrego ●[1], Sten Anslan ●[1], Jeremy R. deWaard[4,9], Stephanie L. deWaard[4,9], Brian L. Fisher ●[5,6], Brendan Furneaux ●[1], Bess Hardwick ●[2], Deirdre Kerdraon[7], Mikko Pentinsaari ●[8], Dimby Raharinjanahary[6], Eric Tsiriniaina Rajoelison[6], Sujeevan Ratnasingham[4], Panu Somervuo ●[2], Jayme E. Sones[4], Evgeny V. Zakharov[4], Paul D. N. Hebert[4], Tomas Roslin ●[2,7] & David Dunson[3]

DNA-based biodiversity surveys result in massive-scale data, including up to millions of species—of which, most are rare. Making the most of such data for inference and prediction requires modeling approaches that can relate species occurrences to environmental and spatial predictors, while incorporating information about their taxonomic or phylogenetic placement. Even if the scalability of joint species distribution models to large communities has greatly advanced, incorporating hundreds of thousands of species has not been feasible to date, leading to compromised analyses. Here we present a 'common to rare transfer learning' (CORAL) approach, based on borrowing information from the common species to enable statistically and computationally efficient modeling of both common and rare species. We illustrate that CORAL leads to much improved prediction and inference in the context of DNA metabarcoding data from Madagascar, comprising 255,188 arthropod species detected in 2,874 samples.

Earth is home to several millions of species[1]. Among these, most are unknown[2] and rare[3]. Innovations in sensor technologies are now providing unprecedented capacity to record patterns and changes in the abundance and distribution of all kinds of taxa, from the named to the previously unnamed and from the rare to the common. These technologies include DNA-based monitoring, passive acoustic monitoring and visual sensors[4,5]. By allowing the efficient recording of thousands to hundreds of thousands of species in time and space, the accumulation

of high-dimensional 'novel community data' is transforming our access to information on species distributions and abundances[4]. As a particularly exciting development, the emergence of novel community data allows us to target the speciose groups accounting for the main part of global biodiversity[1,2]. Where species records so far have been massively biased toward vertebrates, one of the least species-rich taxa[3], recent methods are now making hyper-diverse taxa such as arthropods and fungi arguably easier to sample than vertebrates and plants. As these

[1]Department of Biological and Environmental Science, University of Jyväskylä, Jyväskylä, Finland. [2]Organismal and Evolutionary Biology Research Programme, Faculty of Biological and Environmental Sciences, University of Helsinki, Helsinki, Finland. [3]Department of Statistical Science, Duke University, Durham, NC, USA. [4]Centre for Biodiversity Genomics, University of Guelph, Guelph, Ontario, Canada. [5]Entomology, California Academy of Sciences, San Francisco, CA, USA. [6]Madagascar Biodiversity Center, Parc Botanique et Zoologique de Tsimbazaza, Antananarivo, Madagascar. [7]Department of Ecology, Swedish University of Agricultural Sciences (SLU), Uppsala, Sweden. [8]Canadian National Collection of Insects, Arachnids and Nematodes, Agriculture and Agri-Food Canada, Ottawa, Ontario, Canada. [9]Present address: Smithsonian National Museum of Natural History, Washington, DC, USA. [10]These authors contributed equally: Otso Ovaskainen, Steven Winter. ✉e-mail: otso.t.ovaskainen@jyu.fi

speciose taxa can be mass-sampled and mass-identified, we can derive automated characterizations of what taxa occur where[5–7]. Nonetheless, the revolution in the generation of data is awaiting a matching insurgence of methods to analyze the data.

While most species on Earth are rare, these are the species that we know least about, partially because rare species are the most challenging to model[8]. Paradoxically, the rare species also encompass the taxa in greatest need of protection, and thus the very species for which information on their distributions and ecological requirements is most critical (the rare species paradox[9]). Understanding biodiversity change necessitates models that can relate species occurrences to environmental, biotic and spatial predictors, and which can predict changes in species communities with changes in the state of these drivers[10]. Hence, the need for predictive tools for rare species has been repeatedly highlighted[11–14]. However, the inherent rarity of most species results in highly skewed species-abundance distributions, where a few species are common whereas most species are present at few sites in low numbers. Typical approaches to species-level modeling will then impose a cutoff on species occurrences or abundances[15,16], arguing that for the rarest species, the data are simply insufficient for any quantitative inference regarding the drivers of their distribution. In a world where rare is common[3], this can and will typically amount to rejecting most data, and all the information there then remains hidden. To make the most of increasingly available data, we need modeling approaches that can fully exploit such data.

With species distribution models, rare species may be modeled through ensemble predictions from multiple small models, each of which contains just a few predictors to avoid overfitting[9,17,18]. Because closely related species are generally ecologically more similar than distantly related species[19,20], phylogenetic information may be used to infer the distributions of rare species[21–25]. Joint species distribution models (JSDM)[26] allow leveling up by modeling large numbers of species simultaneously. This enables efficient borrowing of information across species through their shared responses to environmental variation[27]. Furthermore, when data on species phylogenies and/or traits are available, information can be borrowed especially across similar species[28,29]. This can lead to improved predictions, especially for rare species[30].

The high-dimensional, and often extremely sparse, nature of species occurrence data, compounded with spatiotemporal and phylogenetic dependencies, presents major challenges for statistical analyses and computation. High-performance computing can scale some existing JSDMs to thousands of species[31,32]. Two-stage methods, which make small concessions by cutting dependence between species via approximate likelihoods[33,34], can scale to tens of thousands of species while still retaining reasonable uncertainty estimates. Unfortunately, these approaches do not yet scale to the millions of species that comprise the Earth's biodiversity[1]. What is more, they may perform poorly for extremely sparse rare species, by compromising model structures in the interest of gaining computational advantage.

## Results

### The HMSC framework

In this paper, we apply Bayesian transfer learning[35] to develop the common to rare transfer learning (CORAL) approach (Fig. 1). Transfer learning refers to a broad class of multi-stage analysis methods that leverage information from a pretrained model to improve performance for a new but related inference task. In a Bayesian context, this is often achieved by using the posterior model from one dataset to define an intelligent prior model for another dataset. Sharing information between models can improve parameter estimates and substantially boost out-of-sample performance, particularly when studying new, smaller datasets. Our transfer learning method builds on the hierarchical modeling of species communities (HMSC)[10,29,36] approach to joint species distribution modeling[26]. The core idea of CORAL is very general

and will thus apply to many other JSDM approaches, too. What makes its application in the HMSC context so intuitive is that HMSC models species responses to predictors as a function of species traits and phylogenetic relationships. This feature can be efficiently harnessed for transfer learning.

In brief, HMSC is a multivariate generalized linear model fitted in a Bayesian framework. As a response it considers a matrix of species occurrences or abundances. We exemplify our approach with presence–absence data, denoting by $y_{ij} = 1$ if species $j$ (with $j = 1, \dots n_s$) is present in sample $i$ (with $i = 1, \dots n_y$) and $y_{ij} = 0$ if this is not the case. Presence–absence data are modeled in HMSC with probit regression: $\Pr(y_{ij} = 1) = \Phi(L_{ij})$, where $\Phi(.)$ is the standard normal cumulative distribution function and $L_{ij}$ is the linear predictor modeled as:

$$L_{ij} = \sum_{k}^{n_c} x_{ik}\beta_{kj} + \sum_{k}^{n_f} \eta_{ik}\lambda_{kj}, \qquad (1)$$

where $x_{ik}$ are measured predictors, $\eta_{ik}$ are latent predictors, and $\beta_{kj}$ and $\lambda_{kj}$ are regression coefficients quantifying responses of the species to the measured and the latent predictors. The latent features induce within-sample dependence across species; these features may encode characteristics of the habitat, the environment and the spatiotemporal setting not captured by the $x_{ik}$s. HMSC uses a Bayesian hierarchical model to (1) automatically infer how many latent features $n_f$ are needed and (2) to borrow information across species in inferring the $\beta_{kj}$s. For point (2), HMSC estimates to what degree taxonomically or phylogenetically related species, or species with similar traits, show similar responses $\beta_{kj}$ to environmental variation through the multivariate normal distribution[10,28]

$$\mathrm{vec}(B) \sim N(\mu, P \otimes V). \qquad (2)$$

Here, $B$ is the matrix of the regression parameters $\beta_{kj}$ of the $n_s$ species, $\mu$ is the average response, $P = \rho C + (1 - \rho)I n_s$ is a weighted average between the phylogenetic or taxonomic correlation matrix $C$ and the identity matrix $I$ corresponding to unrelated species, $0 \le \rho \le 1$ is the strength of the phylogenetic signal, and $V$ is the variance–covariance matrix of species-specific deviations from the average $\mu$. The average response $\mu$ is further modeled as $\mu = \mathrm{vec}(\Gamma T^T)$, where $T$ is a matrix of species traits, $\Gamma$ are the estimated responses of the traits to environmental variation and the superscript $T$ denotes the matrix transpose. With equation (2), HMSC learns if and to what extent related species, or species with similar traits, show similar environmental responses. This allows for effective borrowing of information among species; for example, improving parameter estimation for rare species, for which it would be difficult to obtain accurate estimates if considering the data in isolation from the community context[29]. As a result, HMSC shows higher predictive performance compared to approaches that do not enable such borrowing of information[30].

### CORAL priors for rare species

Our key idea is that even if it is not feasible to include 100,000+ species in a JSDM model such as HMSC, one can still borrow information from the common species. The structure of our approach follows naturally from three assumptions, namely that (1) users have enough data to perform high-quality inference on common species without leveraging rare species data, (2) information from these common species is relevant for modeling rare species, and (3) rare species may be viewed as conditionally independent given the common species data and measured sample covariates. This suggests a two-stage analysis that first studies the common species jointly and then studies each rare species independently using the results of the common species analysis.

The first stage of CORAL is to fit an HMSC to the common species to pre-estimate latent factors (Fig. 1). From this analysis we obtain a point estimate of the latent features $\eta_i = \eta_{i1}, \dots \eta_{in_f}$), which provides key

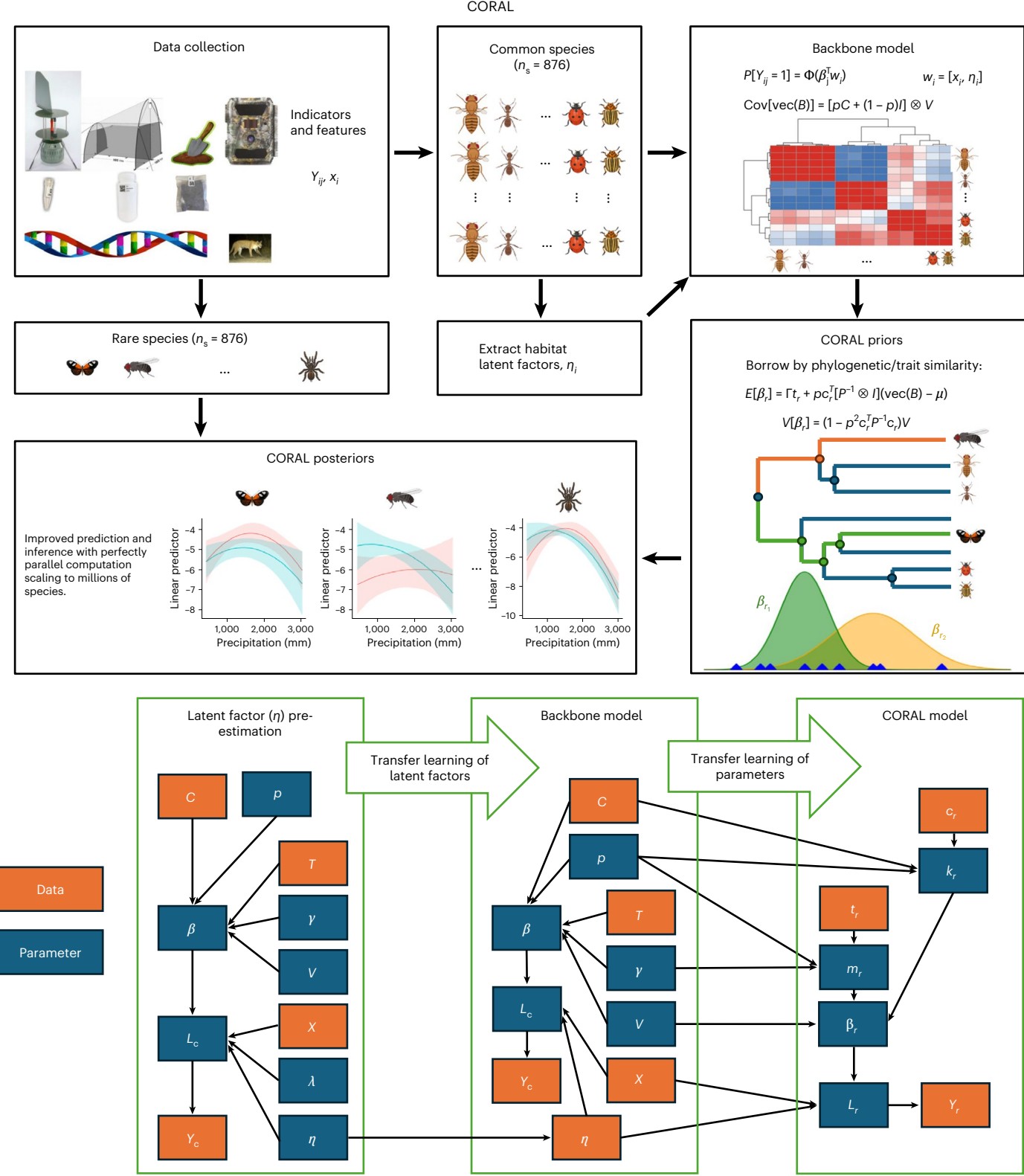

**Fig. 1 | High-level description of the CORAL modeling approach and the generic CORAL model structure.** CORAL is based on fitting a backbone JSDM model to a subset of the most common species in the data and then borrowing information from this backbone to model the rare species. The backbone model learns about latent factors representing relevant missing environmental predictors, as well as about the species responses to both the measured and the latent predictors. The backbone model provides an informative prior distribution for each rare species. This is particularly efficient when we have access to phylogenetic or trait information, allowing information to be borrowed especially from common species closely related to the rare species, or species sharing similar traits with the rare species. CORAL simplifies a fully Bayesian JSDM by replacing latent factors with a pre-estimated point estimate and by accounting for dependence from common to rare species, but not for dependence from rare to common species.

information not captured in $x_i = (x_{i1}, \ldots x_{in_c})$ about environmental and habitat conditions and the overall biological community represented in sample $i$. We define a new covariate vector $\widetilde{\mathbf{x}}_i = \left(\widetilde{x}_{i1}, \ldots, \widetilde{x}_{i(n_c + n_f)}\right)^T$ by concatenating $x_i$ and $\eta_i$ to be used as a fixed predictor in the second stage of CORAL, which fits HMSC to the common species to provide a backbone model (Fig. 1). The third stage fits CORAL models (independent Bayesian probit models) to each rare species: $\Pr(y_{ir} = 1) = \Phi(\widetilde{x}_i^T \beta_r)$, for $r \in \mathcal{J}_r$ with $\mathcal{J}_r \subset \{1, \ldots, n_s\}$ the set of rare species (Fig. 1). To reduce mean square error in inferring $\beta_r$ for $j \in \mathcal{J}_r$, we construct a prior that adaptively shrinks toward the common species coefficients accounting for taxonomic or phylogenetic similarity.

Our prior is motivated by the prior for fixed-effects coefficients in HMSC. To simplify inference and learn relevant hyper-parameters, we first rerun HMSC with the expanded covariate vector $\widetilde{\mathbf{x}}_i$. Under HMSC, the prior conditional distribution of the rare species coefficients given the common species coefficients is

$$\beta_r \sim N(m_r, S_r). \tag{3}$$

Here, the mean is given by

$$m_r = \Gamma \mathbf{t}_r + \left[(\rho c_r^T)(P^{-1} \otimes I_{n_c})\right](\text{vec}(B) - \mu) \tag{4}$$

where $P = \rho C + (1 - \rho) I_{n_s}$. The variance–covariance matrix is $S = k_r V$, where the variance scaling factor $k_r$ is given by

$$k_r = 1 - \rho c_r^T P^{-1} \mathbf{c}_r. \tag{5}$$

The vector $\mathbf{c}_r$ encodes relatedness between a rare species $r \in \mathcal{J}_r$ and all the $n_s$ species in the backbone analysis and $\mathbf{t}_r$ is the trait vector for this rare species. As HMSC is fitted to data with Bayesian inference, parameter uncertainty can be accounted for by defining the prior as a mixture of multivariate normal distributions (each defined by equations (3)–(5)) over the posterior samples. To achieve a simple functional form for the prior, we approximated the mixture by a single multivariate normal distribution, the mean and variance–covariance matrix of which we set equal to those of the mixture.

We refer to the above approach as CORAL. Figure 1 shows the full mathematical structure of this approach, with each box corresponding to a separate stage of CORAL inference. This contrasts with the (computationally intractable) joint modeling approach, which would estimate all parameters for all species simultaneously. CORAL is likely to perform well when assumptions (1)–(3) hold: that is, when there is high-quality common species data that spans the phylogenetic tree and when the backbone model estimates that species responses are phylogenetically structured and/or influenced by species traits. To quantify the benefits of CORAL, we compare its performance to that of a baseline model that does not benefit from the backbone analysis. In other words, for the baseline model we fit $\Pr(y_{ir} = 1) = \Phi(x_i^T \beta_r)$, separately for $r \in \mathcal{J}_r$ using a simple Gaussian prior $\beta_r \sim N(m_r, S_r)$. We expect CORAL to have substantial advantages over the baseline model due to two considerations: $\widetilde{\mathbf{x}}_i$ contains important latent factor information on top of $x_i$, and CORAL allows the borrowing of information from the $\beta_j$s for common species to rare species. As both CORAL and the baseline models can be fitted independently for $r \in \mathcal{J}_r$, computational time scales linearly with the number of species. As a result, these computations can be trivially parallelized allowing for inference and prediction for hundreds of thousands or even millions of species.

To enable easy application of the CORAL approach to high-dimensional biodiversity data, we provide an R package for fitting these models, visualizing the results and generating predictions[37]. This software package also includes a simulated case study that demonstrates how the CORAL approach is able to recover the true parameter values that were used to simulate the data.

## Case study on Malagasy arthropods

We tested the approach in the context of metabarcoding data on Malagasy arthropods. We applied Malaise trap sampling in 53 locations across Madagascar, each of which was relatively undisturbed and where the vegetation represented the conditions of the local environment. We then applied high-throughput cytochrome $c$ oxidase subunit I (COI) metabarcoding[38] and the OptimOTU pipeline[39] to score the occurrences of 255,188 species-level operational taxonomic units (henceforth, species) in 2,874 samples. To create a backbone model of common species, we included those 876 species that occurred in at least 50 samples. This left those 254,312 species that occurred fewer than 50 times in the data as rare species, which we model by the CORAL approach. We note that the threshold of 50 occurrences is relatively high so some of the rare species are not so rare. This choice was made to test the hypothesis that borrowing information from the backbone model changes predictions and inference especially for the very rare species, but less so for more common species. Most of these rare species were extremely rare in the sense that 182,402 species (71% of all rare species) were detected in one sample only. Among these extremely rare species, 1,479 were singletons, that is, represented by a single sequence. Some of these taxa may be artifacts, reflecting chimeric sequences or sequencing error. However, most (99.4%) of the rare species were represented by more than one sequence. Thus, the potential interpretation of some sequencing errors as false species is unlikely to qualitatively influence our conclusions.

As simple and frequently used predictors of species presence, we included covariates related to seasonality, climatic conditions and sequencing effort. Climatic conditions were modeled through the second order polynomials of mean annual temperature and mean annual precipitation[40], while including the interaction between these two climatic predictors. We modeled seasonality through periodic functions $\sin(2\pi d/365)$, $\cos(2\pi d/365)$, $\sin(4\pi d/365)$ and $\cos(4\pi d/365)$, where $d$ is the day of sampling. To capture site-level and sample-level variation not captured by the measured predictors, we included ten site-level ($n = 53$ sites) and four sample-level ($n = 2,874$ samples) latent variables. Variation in sequencing effort was modeled by including log-transformed sequencing depth as a predictor. As a proxy of phylogeny, we used taxonomic assignments at the levels of kingdom, phylum, class, order, family, subfamily, tribe, genus and species, including assignments to pseudotaxa for those cases that could not be reliably classified to previously known taxa.

The common species responded especially to site-level variation (Fig. 2a). This was shown both by responses to climatic variables, which contributed 48% of the explained variation, and by responses to the site-level random factors, which contributed 42% of the explained variation. The effects of the remaining predictors were much less pronounced, with seasonality contributing 3% of the explained variation, sample-level latent factors 7% and sequencing depth 0.1%. As we did not include any traits in the model, we only based the CORAL models on borrowing information on taxonomic relatedness. The responses of the species to the predictors were strongly phylogenetically structured (posterior mean $\rho = 0.65$, posterior probability $\Pr(\rho > 0) = 1.00$), thus providing potential for borrowing information especially from related species.

The variance scaling factor $k$ varied between 0.13 and 0.70, with a mean value of 0.34, thus showing a substantial reduction in variance. As expected, it was lowest for species with close relatives in the backbone model (Fig. 3a). The conditional prior models predicted variation in the occurrences of the rare species better than random (Fig. 3b). This result is nontrivial as the predictions are made by a completely independent model that has not seen any data for the focal species. The accuracy of the prior predictions increased with the level of relatedness between the focal species and the species in the backbone model and the predictions were more accurate for species occurring at least ten times in the data than for the very rare species (Fig. 3b).

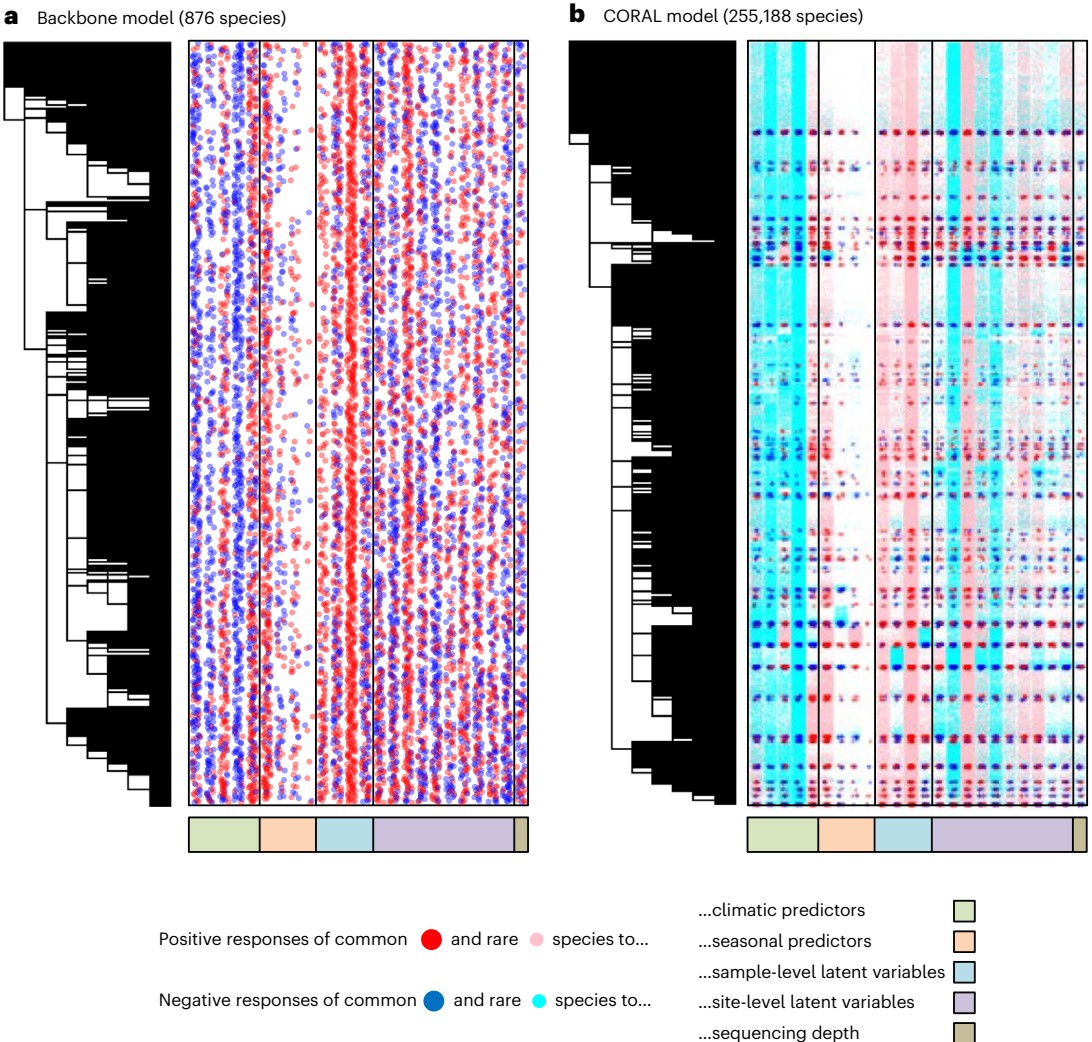

**a** Backbone model (876 species)

**b** CORAL model (255,188 species)

Positive responses of common ● and rare ● species to...

Negative responses of common ● and rare ● species to...

...climatic predictors ▢
...seasonal predictors ▢
...sample-level latent variables ▢
...site-level latent variables ▢
...sequencing depth ▢

**Fig. 2 | Estimated responses of the species to measured and latent predictors. a,b**, The responses are shown for the backbone model of common species (**a**) and for the CORAL model of all species (**b**). Responses that were estimated to be positive (red large dots for common species and pink small dots for rare species) or negative (blue large dots for the common species and cyan small dots for the rare species) with at least 95% posterior probability are highlighted. The dots have been made partially transparent and jittered in the horizontal direction to show the responses to many predictors for a very large number of species.

To compare the baseline and CORAL models in terms of inference, we fitted them to all of the 254,312 rare species. Combining the parameter estimates from the backbone and the CORAL models then enabled us to reveal the responses of all species (common and rare) to the measured and latent predictors (Fig. 2b). These responses illustrate how CORAL transfers information from common species to rare species, as in Fig. 2b blocks of red dots tend to spread pink dots in their surroundings, and blue dots tend to spread cyan dots in their surroundings, meaning that common species induce similar responses to taxonomically related rare species. However, there are exceptions to this general rule, as Fig. 2b shows the CORAL posteriors rather than the CORAL priors. Thus, if the data for a rare species has sufficient evidence of for example positive response even if the related common species show negative responses, the estimate of the rare species will be positive. By updating the conditional prior from the backbone model of common species with data from the focal rare species, we achieved improved predictions in the sense that the CORAL models showed higher precision than the baseline models (Fig. 4). This was especially the case for the very rare species (such as the one exemplified in Fig. 4b), for which the baseline models led to very large credible intervals, as may be expected for the low information contained in few occurrences. For more common species (such as the one exemplified in Fig. 4a), the

increase in precision was smaller (Fig. 4c). The increase in precision increased with relatedness between the focal species and the species included in the backbone model (Fig. 4c), thus mirroring the relation seen between relatedness and the variance scaling factor (Fig. 3a).

One benefit of the CORAL approach is that its posterior distribution is presented analytically rather than through posterior samples obtained through Markov chain Monte Carlo sampling. This is achieved by approximating the CORAL posterior for each species (both the common and the rare) by a multivariate normal distribution. This saves storage space, which could otherwise become limiting for models with very large numbers of species. The multivariate normal presentation of the CORAL posterior also simplifies downstream analyses as posterior mean occurrence probabilities can be computed analytically without Markov chain Monte Carlo sampling. The use of an analytical approximation may, however, introduce model misspecification, the extent of which we explored by comparing the posterior predictive distribution to the data in terms of relevant summary statistics (Fig. 5). The CORAL model fitted to the Malagasy arthropod data was well calibrated in terms of generally predicting the number of times each species was observed, except for some overestimation for the rarest species (Fig. 5a). The model also satisfactorily predicted the number of species present in each sample, but the overestimation in

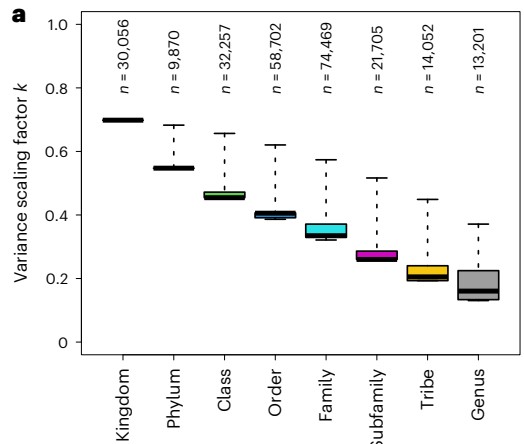
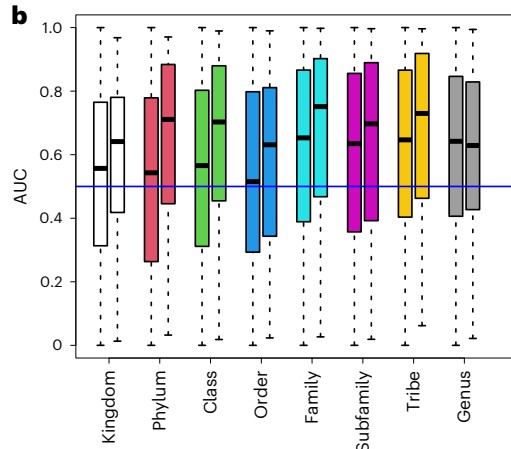

**Fig. 3 | Conditional prior models for rare species constructed by borrowing information from the backbone model of common species. a**, Prior model precision measured by the variance scaling factor $k$ (equation 5), as shown in relation to the taxonomic level shared with the closest relative in the backbone model. The numbers on top of the bars indicate the number of rare species in each category. **b**, Discrimination powers of the conditional prior models, shown separately for each rank of the closest relative in the backbone model (different colors of bars) and for two prevalence classes (at most ten occurrences, left bars; more than ten occurrences, right bars). The blue line shows the null expectation AUC = 0.5. In both panels, the lines show the medians, the boxes the lower and upper quartiles, and the whiskers the minimum and maximum values. In **b**, the numbers of datapoints, (species) included in each box plot are (from left to right) 29,447, 609, 9,725, 145, 31,565, 692, 71,924, 2,545, 20,806, 899, 13,299, 753, 12,477 and 724.

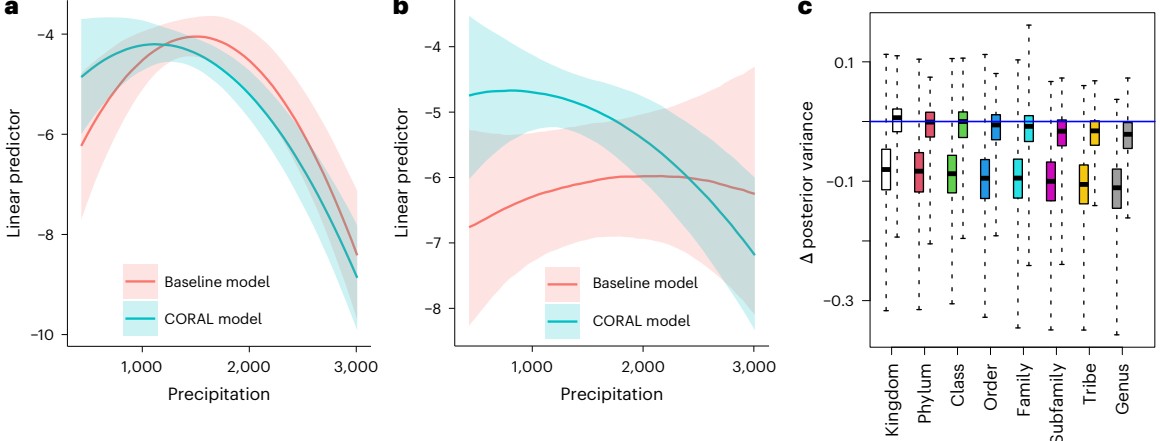

**Fig. 4 | Comparison of inference between CORAL and baseline models.**
**a,b**, A specific prediction for two example species, one that is relatively common (**a**, the wall spider *Garcorops madagascar*, ten occurrences) and another that is very rare (**b**, the deer fly *Chrysops madagascarensis*, one occurrence). The panels show the posterior mean (line) and the interquartile posterior range (shaded area) of the linear predictor under changing precipitation, keeping temperature at its mean value over the data. **c**, Comparison of the posterior variance between baseline and CORAL models systematically for all species. We averaged the posterior variance over the environmental predictors (excluding intercept and latent factors). The panel shows the difference between posterior variance in the CORAL and baseline models. Thus, for values below 0 (the blue line) the CORAL model shows smaller variance. In **c**, the left-hand boxes correspond to very rare species (1–10 occurrences), the right-hand boxes to relatively common species (11–49 occurrences), the lines show the medians, the boxes the lower and upper quartiles, and the whiskers the minimum and maximum values. In **c**, the numbers of datapoints (species) included in each box plot are (from left to right) 29,447, 609, 9,725, 145, 31,565, 692, 71,924, 2,545, 20,806, 899, 13,299, 753, 12,477 and 724.

the occurrences of the rarest species translated to some overestimation of species richness (Fig. 5b). The model fit was uniform across ranges of temperature (Fig. 5c) and humidity (Fig. 5d), suggesting no substantial misspecification in terms of how the effects of these covariates were modeled.

To compare the baseline and CORAL models in terms of predictive power, we considered the 22,140 species that were not included in the backbone model but occurred at least five times in the data. We applied twofold cross validation, where we randomized the folds separately for each species, resampling until both folds included at least 40% of the occurrences. We compared the models using the area under the curve (AUC), Tjur's $R^2$, area under the precision recall curve (PRAUC), Brier score, negative log-likelihood and log-determinant

posterior covariance. Together, these metrics provide a comprehensive overview of model performance covering predictive power, well calibrated probabilities and useful inference. All metrics of predictive performance improved considerably when moving from the baseline model to the CORAL model: AUC from 0.86 to 0.94, Tjur's $R^2$ from 0.03 to 0.08, PRAUC from 0.07 to 0.16, Brier score from 0.004 to 0.003, negative log-likelihood from 0.023 to 0.016 and log determinant from −28.2 to −36.4. All these improvements were significant with $P < 10^{-16}$ as measured by one-sided $t$-tests. The improvement in the predictions was essentially independent of relatedness between the focal species and the species in the backbone model (Fig. 6), suggesting that most of the improvement derived from the inclusion of the latent factors estimated through the joint response of all common species, with less

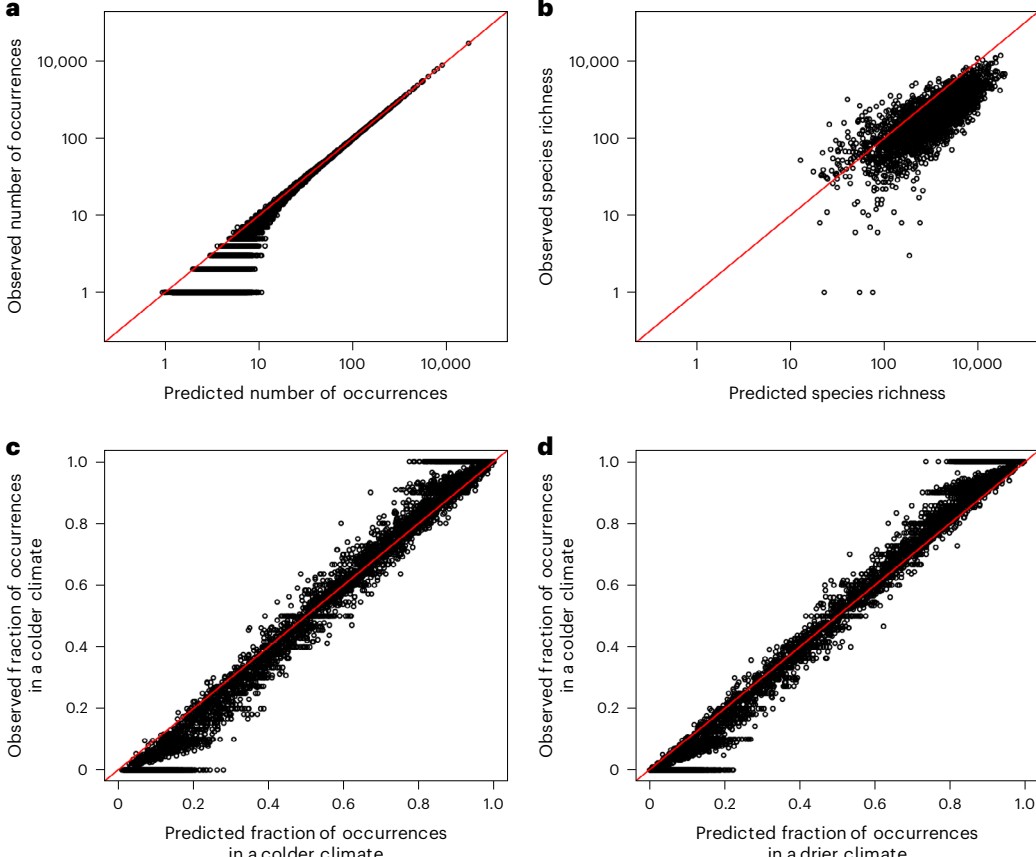

**Fig. 5 | Verification that CORAL posteriors are consistent with the observed data, both in terms of the overall scale of the predicted probabilities as well as the learned covariate effects. a**, The expected number of observations for each species under the CORAL posterior compared to the observed prevalence, with the identity function, is shown as a red line in all figures. **b**, Expected species richness for each sample under the CORAL posterior compared to the observed richness. **c**,**d**, Observed versus predicted proportion of occurrences below the median of temperature (**c**) and precipitation (**d**), shown for species that occur at least ten times in the data.

contribution from the direct borrowing of information from the related species. We validated this inference by fitting another set of models that included common species latent factors but not the CORAL prior; this approach retained about 75% of the gains in AUC over the baseline model. In addition, the mean improvement in AUC did not essentially depend on the prevalence of the species (Fig. 6a), whereas for Tjur's $R^2$ the improvement was higher for the more common species (Fig. 6b).

To further validate the ability of the CORAL approach to infer the environmental responses of the rare species, we rarefied the occurrence data on the common species by masking 90% of their occurrences, thus simulating the case that the common species would actually be rare species. We then used the CORAL approach to estimate the environmental responses of the common species from the rarefied data and compared these estimates to their environmental responses estimated by the backbone model fitted to the full data. The CORAL approach was generally successful in estimating the environmental responses of the species, as there was a high correspondence between the two kinds of estimate (Extended Data Fig. 1): the mean (standard deviation) correlation was 0.68 (0.06) for climatic and seasonal predictors, 0.69 (0.04) for sample-level latent factors and 0.73 (0.04) for site-level latent factors.

## Discussion

The CORAL approach overcomes previous limitations on joint models of species communities with very large numbers of rare species. By borrowing information from a backbone model of common species, CORAL makes it possible to model even the rarest species in a statistically

effective manner by combining an informative prior model with the limited data available for each rare species. As the rare species models can be parameterized independently, CORAL has a perfectly parallel implementation, making it feasible to analyze datasets composed of millions of species. Rather than omitting rare species from all quantitative inference[15,16], the approach developed here enables one to draw on the full information inherent in novel community data[4]. This allows one to generate informed predictions about changes in communities and overall biodiversity with changes in the state of environmental drivers. In essence, this amounts to putting the 'diversity' back in 'biodiversity'.

We found species' responses to climatic, seasonal and latent predictors to be phylogenetically structured (posterior mean $\rho = 0.65$, posterior probability $\Pr (\rho > 0) = 1.00$), forming the basis for borrowing information especially across related species. However, even without phylogenetic signal in the data, or alternatively by fitting a model without phylogeny, CORAL makes it possible to borrow information from the backbone model of common species by identifying sample-level and site-level latent factors, as well as by basing the conditional mean on the average response of all species.

To illustrate the scale of the gain, we reiterate the proportion of rare species in our samples: had we imposed a cutoff of species occurrence in 50 samples, we would have omitted 254,312 out of 255,188 species (99.7%), retaining 876 species (0.3%) of the species pool. Leaving the rare species unmodelled would hardly be an efficient use of the massive data painstakingly acquired. For the 22,140 species (8.7%) that occurred at least five times in the data but were not included in the backbone model, we scored a substantial improvement in predictive

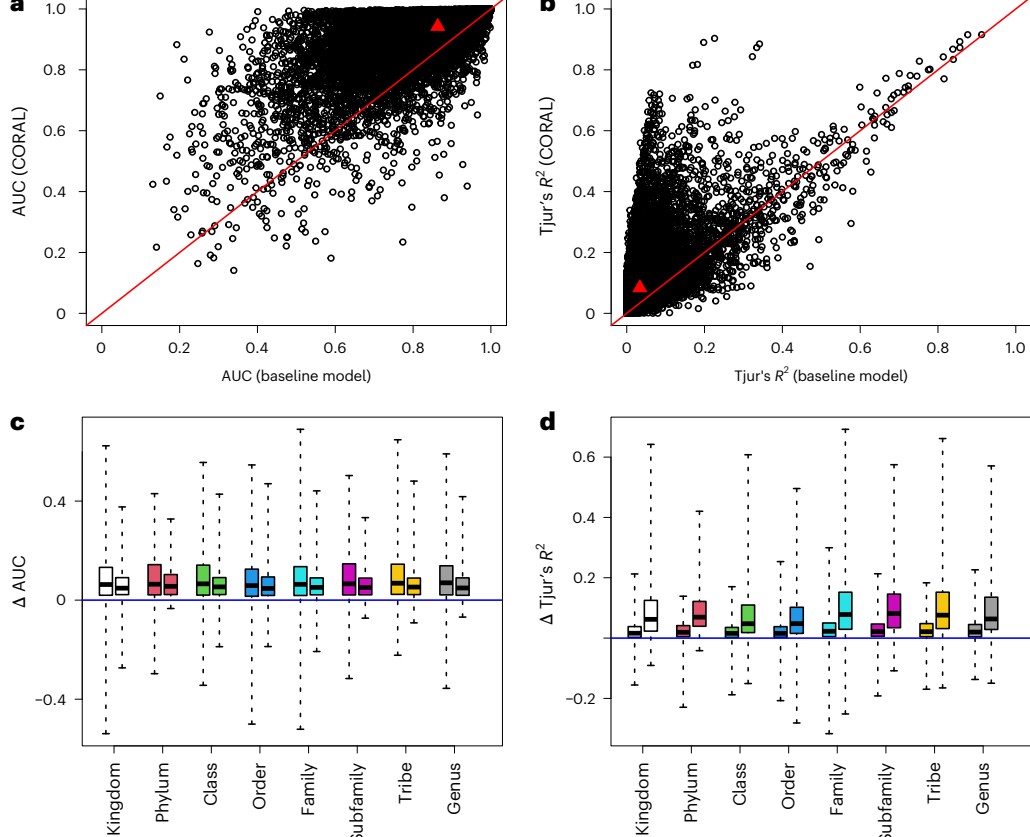

**Fig. 6 | Comparison of predictive power for baseline and CORAL models based on twofold cross validation. a–d**, Predictive comparison is compared in terms of AUC (**a**,**c**) and Tjur's $R^2$ (**b**,**d**). **a**,**b**, The raw values of the metrics for each species included in the analysis, with the red line showing the identity line and the red triangle showing mean values over the species. **c**,**d**, The difference between the CORAL and baseline models. For values above 0 (the blue line), predictions by the CORAL model were more accurate. The results are shown separately for each rank of the closest relative in the backbone model (different colors of bars) and two prevalence classes (ten or fewer occurrences, left bars; more than ten occurrences, right bars). In **c** and **d**, the lines show the medians, the boxes the lower and upper quartiles, and the whiskers the minimum and maximum values. In **c** and **d**, the numbers of datapoints, (species) included in each box plot are (from left to right) 29,447, 609, 9,725, 145, 31,565, 692, 71,924, 2,545, 20,806, 899, 13,299, 753, 12,477 and 724.

power by borrowing information from the more common species. This is a major achievement, as it shows how the limited information inherent in the distribution of rare species may be leveraged by gleaning information from more common species.

While this study focused on methodological development, our findings are also of major interest for understanding the eco-evolutionary community assembly processes of the Malagasy fauna. We found seasonality and climatic responses of arthropods to vary with their phylogenetic relatedness, suggesting that their distributions across Madagascar are partially constrained by their ancestral niche. This region is characterized by extreme levels of endemism at both a regional and a very small scale[41–43]. Nonetheless, in adapting to local conditions, the species appear to maintain a strong signal of their ancestral niche.

## Online content

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

## Methods

### Deriving the CORAL prior

CORAL is motivated by the default prior for coefficients in HMSC. Under this prior, the prior for a species $r$ that is not part of the backbone model (that is, a rare species) is given by $\beta_r | B \sim N(m_r, S_r)$, with the mean (4) and variance (5) given by the conditional multivariate normal formulas. The moments of this distribution are functions of HMSC parameters including $\Gamma$, $\rho$, $V$ and $B$ and do not include information from the common species data, $Y_c$, a priori. Fitting the backbone model produces a posterior distribution, $\pi$, over these parameters, which in turn implies a posterior marginal distribution for the rare species coefficients,

$$p(\beta_r | Y_c) = \int N(\beta_r; m_r, S_r) \, \pi(m_r, S_r | Y_c) \, dm_r dS_r$$

This updated distribution is our desired rare species prior. As this distribution is analytically intractable due to the integral over the posterior; we approximate it with a Gaussian:

$$p(\beta_r | Y_c) \approx N(\beta_r; m'_r, S'_r).$$

The mean $m'_r$ and variance $S'_r$ of this Gaussian are chosen to be the mean and variance of $p(\beta_r | Y_c)$, respectively. These can be calculated using the laws of total expectation and/or variance, resulting in simple expressions in terms of posterior means/variances: $m'_r = E_\pi[m_r]$ and $S'_r = E_\pi[S_r] + V_\pi[m_r]$. In practice, we approximate posterior means and/or variances using Monte Carlo with posterior samples returned by HMSC. This completes specification of the CORAL prior.

### Computational details

We fitted the backbone model with a high-performance computing accelerated version[32] of the R package Hmsc[36], sampling each of the four chains for 37,500 iterations. Of these chains, we omitted the first 12,500 iterations as transient and then thinned the remaining chains by 100 to obtain 250 samples per chain and thus 1,000 posterior samples in total.

For each rare species, we fitted a single-species model where we either did not (the baseline model) or did (the CORAL model) use information from the backbone models of common species. The baseline models were simple probit models with a Gaussian prior on the regression coefficients. The baseline models did not include the latent factors as predictors, and they assumed a default prior distribution for the species responses ($N(0, 10)$ for the intercept and $N(0, 1)$ for fixed effect coefficients). In the CORAL models, we included the latent factors as predictors, and assumed the conditional prior distribution based on equations (3)–(5). We obtained 5000 samples after 2,500 transient iterations for each species for both the baseline and CORAL models using MCMCpack[44].

For each species, we summarized the CORAL model in terms of the mean $\mu$ and variance–covariance matrix $\Sigma$ of the posterior samples. As the model contained 25 parameters (including the intercept), the model for each species was thus represented by $25 + 25(25 + 1)/2 = 350$ parameters (accounting for the symmetry of $\Sigma$). The collection of models for all the 255,188 species thus contained roughly 89 million parameters, which resulted in the manageable file size of around 1.1 GB. We approximate the CORAL posterior through the multivariate normal distribution $N(\mu, \Sigma)$. For predictor vector $\mathbf{x}_i$, the posterior mean of the linear predictor can be then computed as $x_i^T \mu$, and the posterior mean of the occurrence probability as $\Phi(x_i^T \mu / \sqrt{1 + x_i^T \Sigma x_i})$.

### Metrics used to evaluate model performance

AUC is the probability a randomly chosen positive sample has a higher predicted probability than a randomly chosen negative sample. Tjur's $R^2$ is a pseudo-$R^2$ value, which can be read like any other $R^2$ value, but typically reaches lower values[45]. The PRAUC metric quantifies true positives and is useful for analyzing highly imbalanced data where the minority class is of primary interest. The Brier score is the average squared error between predicted probabilities and labels: this metric penalizes overconfidence. Negative log-likelihood directly measures goodness of fit under the proposed Bernoulli model. The determinant of the posterior covariance determines the volume of a 95% credible interval for fixed effect coefficients under a Gaussian approximation, smaller intervals meaning more confident inference. Together, these metrics provide a detailed summary of discrimination ability (AUC, PRAUC), confidence ($R^2$, Brier score), goodness of fit (negative log-likelihood) and precision (log determinant).

### Sampling of Malagasy arthropods

The sampling was conducted as part of the worldwide LIFEPLAN biodiversity sampling design[46]. We selected 53 locations across Madagascar that were relatively undisturbed and where the vegetation represents the conditions of the local environment. Of the sites, 28 were sampled in a spatially nested sampling design with decreasing distances between them (50 km, 5 km and 500 m apart). The other 25 sites were spread across different forested habitats in Madagascar (dry, lowland and montane forests), at elevations ranging from 8 to 1,592 m above sea level. We continuously collected 1-week samples of flying arthropods in 95% ethanol using Malaise traps (ez-Malaise Trap, MegaView Science Co.) Arthropod samples used in this study comply with the regulations for the export and exchange of research samples outlined in the Convention of Biology Diversity and the Convention on International Trade in Endangered Species of Wild Fauna and Flora. Permits to research, collect, and export arthropods were obtained from the Ministry of Environment and Forest as part of an ongoing collaboration between the Ministry of Environment and Forest, Madagascar National Parks, Parc Botanique et Zoologique de Tsimbazaza and the Madagascar Biodiversity Center. Export authorization was provided by the Director of Natural Resources (approval numbers 229/23/MEDD/SG/DGGE/DAPRNE/SCBE.Re). For a detailed description of the sampling, sample shipping and handling, and steps related to DNA extraction and sequencing, we refer to the LIFEPLAN Malaise sample metabarcoding protocol[47].

The CORAL method can be applied to sample $x$ species occurrence data generated by a wide variety of detection technologies and analysis pipelines. In this paper, we applied CORAL to DNA sequence data analyzed using the OptimOTU bioinformatics pipeline[39], which was originally developed for the Global Spore Sampling Project[48] and updated to apply to arthropod COI sequence data as part of the LIFEPLAN biodiversity sampling project[46]. The OptimOTU workflow for COI sequence data consists of primer removal, quality filtering, denoising, de novo and reference-based chimera removal, flagging likely nonanimal sequences, removal of putative nuclear-mitochondrial pseudogenes, probabilistic taxonomic assignment and finally taxonomically guided hierarchical clustering. The OptimOTU pipeline is implemented using the targets v.1.5.1 workflow management package[49], here executed using the crew v.0.9.0 (ref. 50) and crew.cluster v.0.3.0 (ref. 51) backends in R v.4.2.3 (ref. 52) on the Puhti cluster at CSC–IT Center For Science, Finland. This yielded a full taxonomic tree with approximate placeholder taxa to group those sequences that could not be reliably identified.

### Inclusion and ethics statement

The case study on Malagasy arthropods included local researchers (D.R. and E.T.R.) and was conducted in collaboration with a local partner (Madagascar Biodiversity Center). The roles and responsibilities among collaborators were agreed ahead of the research through an Access and Benefit Sharing Agreement.

### Reporting summary

Further information on research design is available in the Nature Portfolio Reporting Summary linked to this article.

## Data availability

The data needed to reproduce the analyses are available via Zenodo at https://doi.org/10.5281/zenodo.11076832 (ref. 37). All raw sequence data are archived on mBRAVE and are publicly available via the European Nucleotide Archive (ENA) at https://www.ebi.ac.uk/ena; project accession number PRJEB86111; run accession numbers ERR15009869– ERR15018787; sample IDs for each accession and download URLs are available via Zenodo at https://doi.org/10.5281/zenodo.11076832 (ref. 37).

## Code availability

The R-based CORAL pipeline needed to reproduce the analyses is available via Zenodo at https://doi.org/10.5281/zenodo.11076832 (ref. 37). The bioinformatics pipeline is available via GitHub at https://github.com/brendanf/CORAL_bioinfo.

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

## Acknowledgements

The work was funded by the European Research Council (ERC) under the European Union's Horizon 2020 research and innovation program (grant 856506: ERC-synergy project LIFEPLAN to O.O., D.D. and T.R.). O.O.'s group was also funded by Academy of Finland (grants 336212 and 345110) and the European Union: HORIZON-CL6-2021-BIODIV-01 project 101059492 (Biodiversity Genomics Europe). J.R.d.W., S.L.d.W., M.P., S.R., J.E.S., E.V.Z. and P.D.N.H. were funded by grants awarded to P.D.N.H. from the Government of Canada through its New Frontiers in Research Fund (grant NFRFT-2020-00073), the Large Scale Applied Research Program administered by Genome Canada and Ontario Genomics (OGI-208) and the Major Science Initiatives Fund administered by Canada Foundation for Innovation (project 42450). We acknowledge CSC—IT Center for Science, Finland, for computational resources.

## Author contributions

O.O. acquired funding, coined the original idea, codeveloped the statistical methodology, performed statistical analyses, contributed to software implementation and cowrote the first draft of the paper. S.W. coined the original idea, codeveloped the statistical methodology, performed statistical analyses, contributed to software implementation and cowrote the first draft of the paper. G.T. codeveloped the statistical methodology, led software implementation and contributed to the first draft of the paper. N.A. contributed to first draft of the paper, in particular placing the statistical framework into ecological context. S.A. contributed to the implementation of the bioinformatics workflow. J.R.d.W. contributed to sample management, DNA extraction and sequencing, and commented on the paper. S.L.d.W. contributed to sample management, DNA extraction and sequencing. B.L.F. acquired funding, participated in project coordination, participated in data collection and commented on the paper. B.F. led the implementation of the bioinformatics workflow and commented on the paper. B.H. participated in project coordination, participated in data collection and commented on the paper. D.K. participated in project coordination and participated in data collection. M.P. contributed to the implementation of the probabilistic taxonomic classification method used in the bioinformatics pipeline. D.R. participated in project coordination and participated in data collection. E.T.R. participated in project coordination and participated in data collection. S.R. contributed to sample management, DNA extraction and sequencing. P.S. led the implementation of the probabilistic taxonomic classification method used in the bioinformatics pipeline and commented on the paper. J.E.S. contributed to sample management, DNA extraction and sequencing. E.V.Z. contributed to sample management, DNA extraction and sequencing. P.D.N.H. contributed to sample management, DNA extraction and sequencing, and commented on the paper. T.R. contributed to first draft of the paper, in particular placing the statistical framework into ecological context. D.D. acquired the funding, coined the original idea, codeveloped the statistical methodology and contributed to the first draft of the paper.

## Funding

## Competing interests

The authors declare no competing interests.

## Additional information

**Extended data** is available for this paper at https://doi.org/10.1038/s41592-025-02823-y.

**Correspondence and requests for materials** should be addressed to Otso Ovaskainen.

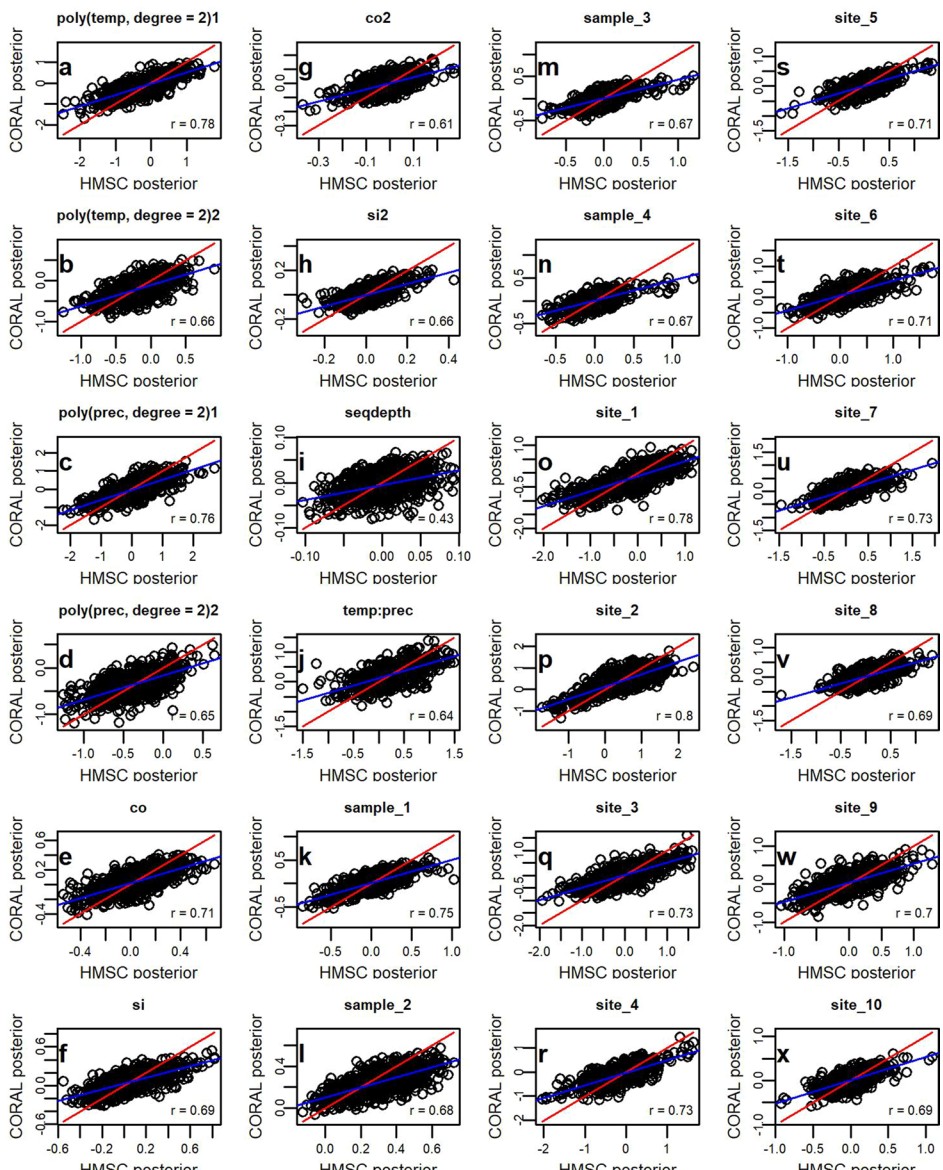

**Extended Data Fig. 1 | Test of CORAL approach's ability to infer the environmental responses of the rare species.** Shown is the correspondence between CORAL (vertical axis) and HMSC (horizontal axis) posteriors for species responses to the fixed effects (**a**–**j**), sample-level latent factors (**k**–**n**), and site-level latent factors (**o**–**x**) in the Madagascar arthropod model. In each panel, the dots correspond to the 876 common species that were included in this analysis. The HMSC posterior is based on fitting the full backbone model to the data. The CORAL posterior is based on applying the CORAL approach to rarefied occurrence data, obtained by masking 90% of occurrences, thus simulating the case that the common species would actually be rare species.

| | |
|---|---|

# Reporting Summary

## Statistics

For all statistical analyses, confirm that the following items are present in the figure legend, table legend, main text, or Methods section.

| n/a | Confirmed | |
|---|---|---|
| ☐ | ☒ | The exact sample size (*n*) for each experimental group/condition, given as a discrete number and unit of measurement |
| ☐ | ☒ | A statement on whether measurements were taken from distinct samples or whether the same sample was measured repeatedly |
| ☐ | ☒ | The statistical test(s) used AND whether they are one- or two-sided *Only common tests should be described solely by name; describe more complex techniques in the Methods section.* |
| ☐ | ☒ | A description of all covariates tested |
| ☐ | ☒ | A description of any assumptions or corrections, such as tests of normality and adjustment for multiple comparisons |
| ☐ | ☒ | A full description of the statistical parameters including central tendency (e.g. means) or other basic estimates (e.g. regression coefficient) AND variation (e.g. standard deviation) or associated estimates of uncertainty (e.g. confidence intervals) |
| ☐ | ☒ | For null hypothesis testing, the test statistic (e.g. *F*, *t*, *r*) with confidence intervals, effect sizes, degrees of freedom and *P* value noted *Give P values as exact values whenever suitable.* |
| ☐ | ☒ | For Bayesian analysis, information on the choice of priors and Markov chain Monte Carlo settings |
| ☐ | ☒ | For hierarchical and complex designs, identification of the appropriate level for tests and full reporting of outcomes |
| ☒ | ☐ | Estimates of effect sizes (e.g. Cohen's *d*, Pearson's *r*), indicating how they were calculated |

*Our web collection on statistics for biologists contains articles on many of the points above.*

## Software and code

Policy information about availability of computer code

| Data collection | No software were used |
|---|---|
| Data analysis | The R-based CORAL pipeline needed to reproduce the analyses is provided at Zenodo: 10.5281/zenodo.11076832 (v4). This pipeline was built using the R-packages phytools (2.1-1), MASS (7.3-60), Hmsc (3.3-3), pROC (1.18.5) and MCMCpack (1.7-0), jsonify (1.2.2), buildmer (2.11), colorspace (2.1-0), matlib (0.9.6), vioplot (0.4.0), MLmetrics (1.1.3) and ggplot2 (3.5.0). The bioinformatics pipeline is available at https://github.com/brendanf/CORAL_bioinfo. |

For manuscripts utilizing custom algorithms or software that are central to the research but not yet described in published literature, software must be made available to editors and reviewers. We strongly encourage code deposition in a community repository (e.g. GitHub). See the Nature Portfolio guidelines for submitting code & software for further information.

## Data

Policy information about availability of data

All manuscripts must include a data availability statement. This statement should provide the following information, where applicable:
- Accession codes, unique identifiers, or web links for publicly available datasets
- A description of any restrictions on data availability
- For clinical datasets or third party data, please ensure that the statement adheres to our policy

The data needed to reproduce the analyses are provided at Zenodo: 10.5281/zenodo.11076832. All raw sequence data are archived on mBRAVE and are publicly

## Human research participants

Policy information about studies involving human research participants and Sex and Gender in Research.

| | |
|---|---|
| Reporting on sex and gender | NA |
| Population characteristics | NA |
| Recruitment | NA |
| Ethics oversight | NA |

Note that full information on the approval of the study protocol must also be provided in the manuscript.

# Field-specific reporting

Please select the one below that is the best fit for your research. If you are not sure, read the appropriate sections before making your selection.

☐ Life sciences       ☐ Behavioural & social sciences       ☒ Ecological, evolutionary & environmental sciences

For a reference copy of the document with all sections, see nature.com/documents/nr-reporting-summary-flat.pdf

# Ecological, evolutionary & environmental sciences study design

All studies must disclose on these points even when the disclosure is negative.

| | |
|---|---|
| Study description | We selected 53 locations across Madagascar that were relatively undisturbed and where the vegetation represents the conditions of the local environment. 28 of the sites were sampled in a spatially nested sampling design with decreasing distances between them (50 km, 5 km and 500 m apart). The other 25 sites were spread across different forested habitats in Madagascar (dry, lowland and montane forests), at elevations ranging from 8 to 1592 MASL. We continuously collected one-week samples of flying arthropods in 95% ethanol using Malaise traps (ez-Malaise Trap, MegaView Science Co.) |
| Research sample | Arthropods collected by Malaise trap over one week period |
| Sampling strategy | Standard Malaise sampling. A detailed sampling protocol is published at https://dx.doi.org/10.17504/protocols.io.kqdg3xkdqg25/v2. |
| Data collection | The data were collected by field teams who were supervised by two coordinators. |
| Timing and spatial scale | The collection started 2021-06-23 and ended 2023-10-11. The 53 collection sites covered essentially the entire Madagascar. |
| Data exclusions | No data were excluded. |
| Reproducibility | The analyses can be replicated with the data and software provided (see data and software availability) |
| Randomization | No randomization was included in data collection. The 53 locations were selected to maximize coverage within Madagascar and based on logistical constraints. |
| Blinding | NA |

Did the study involve field work?       ☒ Yes       ☐ No

## Field work, collection and transport

| | |
|---|---|
| Field conditions | The field conditions were typically to forest research in Madagascar. Mean annual temperature among the sampling sites varies from 17.6 to 27.1 degrees of Celsius, with mean 22.2 degrees of Celsius. |
| Location | Madagascar |
| Access & import/export | Arthropod samples used in this study comply with the regulations for the export and exchange of research samples outlined in the Convention of Biology Diversity and the Convention on International Trade in Endangered Species of Wild Fauna and Flora. Permits to research, collect, and export arthropods were obtained from the Ministry of Environment and Forest as part of an ongoing |

collaboration between the Ministry of Environment and Forest, Madagascar National Parks, Parc Botanique et Zoologique de Tsimbazaza and the Madagascar Biodiversity Center. Export authorization was provided by the Director of Natural Resources. Approval Numbers: 229 /23/MEDD/SG/DGGE/DAPRNE/SCBE.Re.

**Disturbance** | Arthropod traps were set up in natural openings in the forest along existing trails.

# Reporting for specific materials, systems and methods

We require information from authors about some types of materials, experimental systems and methods used in many studies. Here, indicate whether each material, system or method listed is relevant to your study. If you are not sure if a list item applies to your research, read the appropriate section before selecting a response.

## Materials & experimental systems

| n/a | Involved in the study |
|---|---|
| ☒ | ☐ Antibodies |
| ☒ | ☐ Eukaryotic cell lines |
| ☒ | ☐ Palaeontology and archaeology |
| ☐ | ☒ Animals and other organisms |
| ☒ | ☐ Clinical data |
| ☒ | ☐ Dual use research of concern |

## Methods

| n/a | Involved in the study |
|---|---|
| ☒ | ☐ ChIP-seq |
| ☒ | ☐ Flow cytometry |
| ☒ | ☐ MRI-based neuroimaging |

## Antibodies

**Antibodies used** | *Describe all antibodies used in the study; as applicable, provide supplier name, catalog number, clone name, and lot number.*

**Validation** | *Describe the validation of each primary antibody for the species and application, noting any validation statements on the manufacturer's website, relevant citations, antibody profiles in online databases, or data provided in the manuscript.*

## Eukaryotic cell lines

Policy information about cell lines and Sex and Gender in Research

**Cell line source(s)** | *State the source of each cell line used and the sex of all primary cell lines and cells derived from human participants or vertebrate models.*

**Authentication** | *Describe the authentication procedures for each cell line used OR declare that none of the cell lines used were authenticated.*

**Mycoplasma contamination** | *Confirm that all cell lines tested negative for mycoplasma contamination OR describe the results of the testing for mycoplasma contamination OR declare that the cell lines were not tested for mycoplasma contamination.*

**Commonly misidentified lines** (See ICLAC register) | *Name any commonly misidentified cell lines used in the study and provide a rationale for their use.*

## Palaeontology and Archaeology

**Specimen provenance** | *Provide provenance information for specimens and describe permits that were obtained for the work (including the name of the issuing authority, the date of issue, and any identifying information). Permits should encompass collection and, where applicable, export.*

**Specimen deposition** | *Indicate where the specimens have been deposited to permit free access by other researchers.*

**Dating methods** | *If new dates are provided, describe how they were obtained (e.g. collection, storage, sample pretreatment and measurement), where they were obtained (i.e. lab name), the calibration program and the protocol for quality assurance OR state that no new dates are provided.*

☐ Tick this box to confirm that the raw and calibrated dates are available in the paper or in Supplementary Information.

**Ethics oversight** | *Identify the organization(s) that approved or provided guidance on the study protocol, OR state that no ethical approval or guidance was required and explain why not.*

Note that full information on the approval of the study protocol must also be provided in the manuscript.

# Animals and other research organisms

Policy information about studies involving animals; ARRIVE guidelines recommended for reporting animal research, and Sex and Gender in Research

| Laboratory animals | The study did not involve laboratory animals. |
|---|---|
| Wild animals | The study involves Malaise-trapping of arthropods. We continuously collected one-week samples of flying arthropods in 95% ethanol using Malaise traps. The animals died in the ethanol. The number of different species is ca. 250,000. We could not age nor sex the individuals, the study focuses on species-level classification. Most species remain unknown for science and cannot thus be named. |
| Reporting on sex | We could not sex the individuals, the study focuses on species-level classification. |
| Field-collected samples | We collected samples of flying arthropods in 95% ethanol using Malaise traps. The study did not include live animals collected from the field. |
| Ethics oversight | No ethical approval is required with arthropod sampling. |

Note that full information on the approval of the study protocol must also be provided in the manuscript.

# Clinical data

Policy information about clinical studies
All manuscripts should comply with the ICMJE guidelines for publication of clinical research and a completed CONSORT checklist must be included with all submissions.

| Clinical trial registration | *Provide the trial registration number from ClinicalTrials.gov or an equivalent agency.* |
|---|---|
| Study protocol | *Note where the full trial protocol can be accessed OR if not available, explain why.* |
| Data collection | *Describe the settings and locales of data collection, noting the time periods of recruitment and data collection.* |
| Outcomes | *Describe how you pre-defined primary and secondary outcome measures and how you assessed these measures.* |

# Dual use research of concern

Policy information about dual use research of concern

## Hazards

Could the accidental, deliberate or reckless misuse of agents or technologies generated in the work, or the application of information presented in the manuscript, pose a threat to:

| No | Yes | |
|---|---|---|
| ☒ | ☐ | Public health |
| ☒ | ☐ | National security |
| ☒ | ☐ | Crops and/or livestock |
| ☒ | ☐ | Ecosystems |
| ☒ | ☐ | Any other significant area |

## Experiments of concern

Does the work involve any of these experiments of concern:

| No | Yes | |
|---|---|---|
| ☒ | ☐ | Demonstrate how to render a vaccine ineffective |
| ☒ | ☐ | Confer resistance to therapeutically useful antibiotics or antiviral agents |
| ☒ | ☐ | Enhance the virulence of a pathogen or render a nonpathogen virulent |
| ☒ | ☐ | Increase transmissibility of a pathogen |
| ☒ | ☐ | Alter the host range of a pathogen |
| ☒ | ☐ | Enable evasion of diagnostic/detection modalities |
| ☒ | ☐ | Enable the weaponization of a biological agent or toxin |
| ☒ | ☐ | Any other potentially harmful combination of experiments and agents |

# ChIP-seq

## Data deposition

☐ Confirm that both raw and final processed data have been deposited in a public database such as GEO.

☐ Confirm that you have deposited or provided access to graph files (e.g. BED files) for the called peaks.

| | |
|---|---|
| **Data access links**<br>*May remain private before publication.* | *For "Initial submission" or "Revised version" documents, provide reviewer access links. For your "Final submission" document, provide a link to the deposited data.* |
| **Files in database submission** | *Provide a list of all files available in the database submission.* |
| **Genome browser session**<br>(e.g. UCSC) | *Provide a link to an anonymized genome browser session for "Initial submission" and "Revised version" documents only, to enable peer review. Write "no longer applicable" for "Final submission" documents.* |

## Methodology

| | |
|---|---|
| **Replicates** | *Describe the experimental replicates, specifying number, type and replicate agreement.* |
| **Sequencing depth** | *Describe the sequencing depth for each experiment, providing the total number of reads, uniquely mapped reads, length of reads and whether they were paired- or single-end.* |
| **Antibodies** | *Describe the antibodies used for the ChIP-seq experiments; as applicable, provide supplier name, catalog number, clone name, and lot number.* |
| **Peak calling parameters** | *Specify the command line program and parameters used for read mapping and peak calling, including the ChIP, control and index files used.* |
| **Data quality** | *Describe the methods used to ensure data quality in full detail, including how many peaks are at FDR 5% and above 5-fold enrichment.* |
| **Software** | *Describe the software used to collect and analyze the ChIP-seq data. For custom code that has been deposited into a community repository, provide accession details.* |

# Flow Cytometry

## Plots

Confirm that:

☐ The axis labels state the marker and fluorochrome used (e.g. CD4-FITC).

☐ The axis scales are clearly visible. Include numbers along axes only for bottom left plot of group (a 'group' is an analysis of identical markers).

☐ All plots are contour plots with outliers or pseudocolor plots.

☐ A numerical value for number of cells or percentage (with statistics) is provided.

## Methodology

| | |
|---|---|
| **Sample preparation** | *Describe the sample preparation, detailing the biological source of the cells and any tissue processing steps used.* |
| **Instrument** | *Identify the instrument used for data collection, specifying make and model number.* |
| **Software** | *Describe the software used to collect and analyze the flow cytometry data. For custom code that has been deposited into a community repository, provide accession details.* |
| **Cell population abundance** | *Describe the abundance of the relevant cell populations within post-sort fractions, providing details on the purity of the samples and how it was determined.* |
| **Gating strategy** | *Describe the gating strategy used for all relevant experiments, specifying the preliminary FSC/SSC gates of the starting cell population, indicating where boundaries between "positive" and "negative" staining cell populations are defined.* |

☐ Tick this box to confirm that a figure exemplifying the gating strategy is provided in the Supplementary Information.

# Magnetic resonance imaging

## Experimental design

| | |
|---|---|
| **Design type** | *Indicate task or resting state; event-related or block design.* |

| Design specifications | *Specify the number of blocks, trials or experimental units per session and/or subject, and specify the length of each trial or block (if trials are blocked) and interval between trials.* |
|---|---|
| Behavioral performance measures | *State number and/or type of variables recorded (e.g. correct button press, response time) and what statistics were used to establish that the subjects were performing the task as expected (e.g. mean, range, and/or standard deviation across subjects).* |

## Acquisition

| Imaging type(s) | *Specify: functional, structural, diffusion, perfusion.* |
|---|---|
| Field strength | *Specify in Tesla* |
| Sequence & imaging parameters | *Specify the pulse sequence type (gradient echo, spin echo, etc.), imaging type (EPI, spiral, etc.), field of view, matrix size, slice thickness, orientation and TE/TR/flip angle.* |
| Area of acquisition | *State whether a whole brain scan was used OR define the area of acquisition, describing how the region was determined.* |

Diffusion MRI ☐ Used ☐ Not used

## Preprocessing

| Preprocessing software | *Provide detail on software version and revision number and on specific parameters (model/functions, brain extraction, segmentation, smoothing kernel size, etc.).* |
|---|---|
| Normalization | *If data were normalized/standardized, describe the approach(es): specify linear or non-linear and define image types used for transformation OR indicate that data were not normalized and explain rationale for lack of normalization.* |
| Normalization template | *Describe the template used for normalization/transformation, specifying subject space or group standardized space (e.g. original Talairach, MNI305, ICBM152) OR indicate that the data were not normalized.* |
| Noise and artifact removal | *Describe your procedure(s) for artifact and structured noise removal, specifying motion parameters, tissue signals and physiological signals (heart rate, respiration).* |
| Volume censoring | *Define your software and/or method and criteria for volume censoring, and state the extent of such censoring.* |

## Statistical modeling & inference

| Model type and settings | *Specify type (mass univariate, multivariate, RSA, predictive, etc.) and describe essential details of the model at the first and second levels (e.g. fixed, random or mixed effects; drift or auto-correlation).* |
|---|---|
| Effect(s) tested | *Define precise effect in terms of the task or stimulus conditions instead of psychological concepts and indicate whether ANOVA or factorial designs were used.* |

Specify type of analysis: ☐ Whole brain ☐ ROI-based ☐ Both

| Statistic type for inference (See Eklund et al. 2016) | *Specify voxel-wise or cluster-wise and report all relevant parameters for cluster-wise methods.* |
|---|---|
| Correction | *Describe the type of correction and how it is obtained for multiple comparisons (e.g. FWE, FDR, permutation or Monte Carlo).* |

## Models & analysis

n/a | Involved in the study
☐ ☐ Functional and/or effective connectivity
☐ ☐ Graph analysis
☐ ☐ Multivariate modeling or predictive analysis

| Functional and/or effective connectivity | *Report the measures of dependence used and the model details (e.g. Pearson correlation, partial correlation, mutual information).* |
|---|---|
| Graph analysis | *Report the dependent variable and connectivity measure, specifying weighted graph or binarized graph, subject- or group-level, and the global and/or node summaries used (e.g. clustering coefficient, efficiency, etc.).* |
| Multivariate modeling and predictive analysis | *Specify independent variables, features extraction and dimension reduction, model, training and evaluation metrics.* |

