## [Peer Review File · Nature Methods]

Common to rare transfer learning (CORAL) enables inference and prediction for a quarter million rare Malagasy arthropods

Corresponding Author: Professor Otso Ovaskainen

Version 0:

Decision Letter:

23rd Jan 2025

Dear Professor Ovaskainen,

I am very sorry about the excessive delays in the review process. However, I have some good news.

Your Article, "Common to rare transfer learning (CORAL) enables inference and prediction for a quarter million rare Malagasy arthropods", has now been seen by three reviewers. As you will see from their comments below, although the reviewers find your work of considerable potential interest, they have raised a number of concerns. We are interested in the possibility of publishing your paper in Nature Methods, but would like to consider your response to these concerns before we reach a final decision on publication.

We therefore invite you to revise your manuscript to address these concerns. Please make sure to provide a user-friendly software implementation of your approach. Please also make sure that the manuscript is understandable to a general audience.

Link Redacted

We hope to receive your revised paper within two months. If you cannot send it within this time, please let us know. In this event, we will still be happy to reconsider your paper at a later date so long as nothing similar has been accepted for publication at Nature Methods or published elsewhere.

OPEN SCIENCE REQUIREMENTS

REPORTING SUMMARY AND EDITORIAL POLICY CHECKLISTS

EXTENDED DATA FIGURES

DATA AVAILABILITY

All novel DNA and RNA sequencing data, protein sequences, genetic polymorphisms, linked genotype and phenotype data, gene expression data, macromolecular structures, and proteomics data must be deposited in a publicly accessible database, and accession codes and associated hyperlinks must be provided in the "Data Availability" section.

CODE AVAILABILITY

Please include a “Code Availability” subsection in the Online Methods which details how your custom code is made available. Only in rare cases (where code is not central to the main conclusions of the paper) is the statement “available upon request” allowed (and reasons should be specified).

MATERIALS AVAILABILITY

ORCID

Nature Methods is committed to improving transparency in authorship. As part of our efforts in this direction, we are now requesting that all authors identified as ‘corresponding author’ on published papers create and link their Open Researcher and Contributor Identifier (ORCID) with their account on the Manuscript Tracking System (MTS), prior to acceptance. This applies to primary research papers only. ORCID helps the scientific community achieve unambiguous attribution of all scholarly contributions. You can create and link your ORCID from the home page of the MTS by clicking on ‘Modify my Springer Nature account’. For more information please visit www.springernature.com/orcid.

Best regards,
Nina

Nina Vogt, PhD
Senior Editor
Nature Methods

Reviewers' Comments:

Reviewer #1 (Remarks to the Author):

Comments for the authors

In this paper, the authors proposed a “common to rare transfer learning” (CORAL) method to model joint species distribution under a hierarchical Bayesian framework. The method was illustrated using a large DNA metabarcoding data on Madagascar arthropods. The CORAL approach is based on fitting a backbone JSDM (Joint Species Distribution Modeling) method to a subset of the common species in the data. The information from this backbone model is then used to make inferences and predictions about the rare species. I have some questions regarding the data analysis and the statistical assumptions underlying the CORAL method.

(1) Statistical assumptions and interpretations

(a) Although the basic idea of using the backbone model for abundant species to model rare species is clearly presented in Figure 1 of the manuscript, the assumptions and priors underlying the CORAL model may require further explanation or justification. Unless readers have a background or sufficient knowledge of joint species distribution modeling under a Bayesian framework, the CORAL model and its underlying assumptions may not be easily understood and implemented by a general audience.

(b) The interpretation of the model should be strengthened. For example, the measured predictors include temperature, temperature squared, precipitation, precipitation squared and the interaction of temperature and precipitation (Figure 2).

However, in Figure 5a and 5b, only the effects of precipitation are shown. Readers would likely be more interested in understanding how temperature and precipitation and their interaction affect the occurrence probability of each species.

(c) The authors indicated (L187) that the CORAL model included site ($n=53$) and sample ($n=2874$) as random effects. What are the estimated variances for these two random effects? It would be helpful to provide proper interpretations and implications regarding these random effects. Additionally, the authors may need to clarify why there are responses for only 4 (latent?) samples and 10 (latent?) sites in Figure 2.

(d) In Figure 5a, the “linear predictor” (on the Y-axis) takes values between -10 and -4. What are the corresponding occurrence probabilities under a probit regression? What would the occurrence probabilities be if the probit regression were replaced by a logistic or other types of regression?

(e) There are several other model assumptions that need to be addressed. The covariance matrix of $\text{vec}(\beta)$ and CORAL priors also require justification. Although the covariance matrix of $\text{vec}(\beta)$ and the CORAL priors are convenient and typical model assumptions in a Bayesian framework, I wonder how robust the CORAL model is when the covariance matrix and priors are mis-specified?

(2) Metabarcoding data are often subject to sequencing errors

The authors applied high-throughput COI metabarcoding and the OptimOTU pipeline to record the occurrences of 255,188 OTUs (“species”) in 2874 samples collected in 53 locations across Madagascar. The authors treated those species that occurred in at least 50 samples as “abundant” (there were 876 of them) and the other 254,312 species as “rare”. However, it is well known that sequencing errors often generate spurious low frequency counts, especially singletons, which can lead to an inflated number of rare species and incorrect estimates of diversity (Edgar, 2013; 2017; Bunge et al. 2014). For example, sequencing errors can often cause two sequences that should be classified into the same OTU to be classified as distinct OTUs or as singletons if no match is found. The authors indicated (Line 257 of the manuscript) that there were 182,402 (~71% of all species) singletons, which were detected only once in the OTU data. I wonder whether sequencing errors exist in the authors’ data. If a portion of the singletons are spurious, it would not be meaningful to make inferences about species with only one occurrence. To my knowledge, there are at least two statistical methods to “adjust” the number of singletons: Edgar (2018) and Chiu and Chao (2019). These adjusted methods provide estimates of true singletons, which can then be used to assess sample completeness; see Point (3) below.

(3) Assessment of sample completeness

Based on Alan Turing’s statistical perspective, the existence of a large number of singletons implies that many species remain undetected in the data (Good 1953). According to Good (1953), this concept was originally developed by Turing in his cryptographic analysis during World War II. For hyper-diverse arthropod data, it would be useful to quantify sample completeness before any joint species modeling. Conventional sample completeness is defined as the proportion of observed species richness to true richness (observed plus undetected species). However, data are often insufficient to provide an accurate estimate of true species richness. In contrast, Turing’s objective measure of sample completeness is based on the concept of sample coverage (the fraction of an assemblage’s individuals that belong to the observed species). Contrary to most people’s intuition, sample coverage can be accurately estimated based on the sample data itself; Turing’s estimator (one minus the proportion of singletons) is surprisingly simple. For Metabarcoding data, after adjusting for the number of singletons, one can accurately quantify sample completeness. See Slik et al. (2015) for an example.

References used in the review:

Bunge et. al. (2014). Estimating the number of species in microbial diversity studies. Annual Review of Statistics and Its Application, 1, 427-445.

Chiu, C-H. and Chao, A. (2016). Estimating and comparing microbial diversity in the presence of sequencing errors. PeerJ 4:e1634.

Edgar R. C. (2013). UPARSE: highly accurate OTU sequences from microbial amplicon reads. Nature Methods 10: 996–998.

Edgar R. C. (2017). Accuracy of microbial community diversity estimated by closed- and open reference OTUs. PeerJ. doi: 10.7717/peerj.3889.

Edgar R. C. and Flyvbjerg, H. (2018) Alpha diversity metrics for noisy OTUs. BioRxiv <https://www.biorxiv.org/content/10.1101/434977v1>

Good I. J. (1953) The population frequencies of species and the estimation of population parameters. Biometrika, 40, 237-264.

Slik, J. W. F. et al. (2015) An estimate of the number of tropical tree species. PNAS 112, 7472-7477.

Reviewer #2 (Remarks to the Author):

This paper proposes a new approach (CORAL) for simultaneous inference across multiple species using transfer learning. Important challenges in joint species distribution modelling (JSDM) are handling many species and constructing models for rare species. The proposed approach addresses these by first fitting a "backbone model" to a smaller number of abundant species, then fitting parallel models to rare species using a transfer learning approach, where the model is simplified by taking observed values for latent variables from the backbone model. The method is shown to work very well on a Madagascan arthropod dataset of hundreds of thousands of species, dramatically improving predictive performance. This is an original and useful idea which could have widespread impact in the field. I have no major concerns and comment mostly on the exposition.

CORAL is proposed as an extension of HMSC, a specific software application for fitting JSDMs, but the method is quite general and could be used with other software too. It would have been nice to see some discussion of this, or a formulation of the problem that was not HMSC-specific.

Some guidance for users would be helpful - is there a software implementation of this approach? If not, is there some simple example code you could share for how to implement this approach?

The Methods section only describes the computational methods used in very general terms, while going into a lot of detail about the experimentation and bioinformatics pipeline used to collect and pre-process the data. More information about the statistical procedures used is needed, especially given that this is the main focus of the paper.

The idea of transfer learning is key to the paper but it is not explained to readers at all, only a reference is given to a paper on Bayesian transfer learning. It would be helpful to provide some context here, and explanation of how transfer learning works, when it was proposed, where it is typically used.

It is stated a couple of times (lines 83, 285) that because the proposed approach is scalable to millions of species it is capable of modelling all of Earth's biodiversity... this is an attractive idea but I feel like there is some overreach here. This approach only deals with a large number of responses (species) not a large number of observations (sites/replicates). To model all of Earth's biodiversity we would need a pretty large sample size, in order to capture the diversity of environments in which we find species, and there would be further technical challenges to overcome in doing this (including capturing spatial structure, which would vary at a relatively fine scale in a global analysis). Perhaps some minor rephrasing could clear this up.

Reviewer #2 (Remarks on code availability):

I partly reviewed the code - I looked at the demo simulated dataset - and was confused as to why the second stage backbone model fit (which I believe is predicting rare species, using the backbone model constructed from abundant species?) happened before calculation of the prior variance for rare species? Don't you need the prior variance in order to fit the rare species model?

Perhaps the workflow for implementing CORAL could be clarified...

Reviewer #3 (Remarks to the Author):

This paper develops and tests a statistical framework to predict the distribution of species based on environmental predictors. One problem for this kind of species distribution models is that rare species often do not have enough data to allow robust prediction. This paper addresses the issue by using phylogenetic relationships between species to infer the environmental parameters of the rare species, through a bayesian statistical framework. I think that the approach is novel and has an interesting potential. But I have some major concerns about the current draft of the paper as I detail below.

General Comments

1. The statistical approach used by the model is arguably very complex, and I had difficulty in following it at several times. I found that some of the explanations could be made more understandable. In addition it is not clear that the complexity is always warranted. For instance, the authors attempt to make the approach as generic as possible to also allow for the use of traits. But then they do not use the trait matrix in the application of the model, so not only we don't know if it can be effective, it adds a layer of complexity that is not ultimately needed in this paper.

2. A second issue related to the complexity of the approach is the issue of reusability of this approach. We are provided with some R code (which I did not have time to examine), but it seems to imply access to massive computational power. I would like to have seen a small version of the model/test dataset for the less resourced scientists to examine.

3. Test of the approach. I found it challenging to assess the testing of the approach. I am not sure I understand what is shown in Figures 3 and 4 and the range of metrics chosen. In addition I would like to have seen a test of the model with data for which we had a better way of testing the model fit. For instance, one could have a subset of common species for which a subsample would be generated. We know for those species what are the predicted occurrences and what are the important

predictor variables. Is this inference approach based on phylogenetics good at picking those predictor variables? Alternatively a synthetic dataset would also allow us to assess the model fit for known/controlled conditions.

Specific Comments

I.38. We don't know how many species there are in the planet. Most estimates suggest between 4 and 10 Million, but given a single number is misleading.

I.111. The use of T to represent the matrix of traits and the transpose operation is very confusing. Suggest use "t" for transpose or change the letter for the trait matrix.

I.106, 111. Why are B and mu vectorised? Is this for computational purpose or analytical purposes?

I.183-184. The choice of the climate predictors seems a bit limited (annual values only) and why was a semestral seasonality added to the annual seasonality?

Figure 2. Why are there only 4 samples and 10 sites? Is this a subset of the study?

Version 1:

Decision Letter:

Our ref: NMETH-A56224A

21st Apr 2025

Dear Otso,

Thank you for submitting your revised manuscript "Common to rare transfer learning (CORAL) enables inference and prediction for a quarter million rare Malagasy arthropods" (NMETH-A56224A). It has now been seen by the original referees and their comments are below. The reviewers find that the paper has improved in revision, and therefore we'll be happy in principle to publish it in Nature Methods, pending minor revisions to satisfy the referees' final requests and to comply with our editorial and formatting guidelines.

Please do include the new data you have sent in response to my previous email. Please also make sure that the Zenodo deposition is clearly described.

TRANSPARENT PEER REVIEW

ORCID

Best regards,
Nina

Nina Vogt, PhD
Senior Editor

Reviewer #1 (Remarks to the Author):

In the revision, the authors have properly responded or implemented my questions/comments raised in my earlier review. I have no further comments. I

Reviewer #1 (Remarks on code availability):

None

Reviewer #2 (Remarks to the Author):

I have reviewed the author response, the changes to the manuscript in response to reviewer comments, and some of the new demo code demonstrating how to apply CORAL in practice. I think the authors have done an excellent job responding to reviewer comments and have no further suggestions.

Reviewer #3 (Remarks to the Author):

I thank the authors for addressing some of my comments in a previous draft. However I don't think they addressed fully one of my major comments regarding the testing of the approach. The authors just argue they are using the ideal approach to test the data. I don't think that predicting the occurrence of rare species better than by random is the "ideal" approach (random seems like a low bar to me, but maybe I am missing something). My proposal was to use a subsetting of the data for the test, a bit like in a training/validation split, but using "rarified" occurrences from the common species data. They say they provided "a software demonstration with simulated data" that showed "the validity of the approach for known conditions" but fail to point out exactly in what page of the manuscript this is done. So I would still like to see this issue better addressed or explained.

Manuscript ID: NMETH-A56224

Title: Common to rare transfer learning (CORAL) enables inference and prediction for a quarter million rare Malagasy arthropods

Journal: *Nature Methods*

Dear Senior Editor Nina Vogt, PhD

Thank you very much for the careful assessment of our manuscript and for inviting us to submit a revised version. We have now responded to all comments by the three reviewers, which we feel have greatly improved the clarity of our work. In particular, they have helped us make the manuscript understandable to a general audience and to generate a user-friendly software implementation of our approach. As a result of the modifications, we feel that the manuscript has improved considerably, and we thank the reviewers for their constructive comments.

Below, we provide a point-by-point response to all the comments. The original comments are shown in *italics* and our responses in regular font and starting with ">>>". In the main text, the implemented revisions are highlighted with **red font**.

We trust you will find the current version of the manuscript publishable in *Nature Methods*.

Kind regards,

Prof. Otso Ovaskainen and co-authors.

Reviewer #1

In this paper, the authors proposed a “common to rare transfer learning” (CORAL) method to model joint species distribution under a hierarchical Bayesian framework. The method was illustrated using a large DNA metabarcoding data on Madagascar arthropods. The CORAL approach is based on fitting a backbone JSJM (Joint Species Distribution Modeling) method to a subset of the common species in the data. The information from this backbone model is then used to make inferences and predictions about the rare species. I have some questions regarding the data analysis and the statistical assumptions underlying the CORAL method.

>>> We thank the reviewer for the accurate summary of our work, and for questions and comments which have helped us improve our presentation.

(1) Statistical assumptions and interpretations

(a) Although the basic idea of using the backbone model for abundant species to model rare species is clearly presented in Figure 1 of the manuscript, the assumptions and priors underlying the CORAL model may require further explanation or justification. Unless readers have a background or sufficient knowledge of joint species distribution modeling under a Bayesian framework, the CORAL model and its underlying assumptions may not be easily understood and implemented by a general audience.

>>> We fully agree. To improve clarity, we have now significantly expanded the CORAL section. In particular, we include an updated version of Figure 1 which shows the mathematical structure of CORAL relative to the computationally intractable full model. We now clearly explain what assumptions underlie CORAL and when these assumptions are likely to hold.

(b) The interpretation of the model should be strengthened. For example, the measured predictors include temperature, temperature squared, precipitation, precipitation squared and the interaction of temperature and precipitation (Figure 2). However, in Figure 5a and 5b, only the effects of precipitation are shown. Readers would likely be more interested in understanding how temperature and precipitation and their interaction affect the occurrence probability of each species.

>>> In Fig. 5, we have explicitly chosen to illustrate responses to a few predictors only. This choice is required to achieve a quantitative illustration of the amount of uncertainty in these predictions. However, we agree with the Reviewer that we had previously failed to provide a comprehensive picture of how the rare species responded to the predictors. How the common species included in the backbone model respond to the predictors was shown in Fig. 2 of the original submission. To include the rare species, we have now expanded Fig. 2 to include a comparable illustration (new Fig. 2B) showing how *all* species (both the common and rare ones) respond to the predictors. As we expect most applicants of our method will be interested in this very feature, we find that the addition of this new illustration has greatly strengthened the take-home message of our work.

(c) The authors indicated (L187) that the CORAL model included site ($n=53$) and sample ($n=2874$) as random effects. What are the estimated variances for these two random effects? It would be helpful to provide proper interpretations and implications regarding these random effects. Additionally, the authors may need to clarify why there are responses for only 4 (latent?) samples and 10 (latent?) sites in Figure 2.

>>> One key benefit of HMSC and CORAL is that they utilize a Bayesian approach which automatically selects the number of latent factors. In our analysis, this results in 10 site-level latent factors which explain 42% of the variation in the data, and in 4 sample-level random factors which explain another 7%. To make this clearer, we have now updated Figure 2 and the HMSC section.

(d) In Figure 5a, the “linear predictor” (on the Y-axis) takes values between -10 and -4. What are the corresponding occurrence probabilities under a probit regression? What would the occurrence probabilities be if the probit regression were replaced by a logistic or other types of regression?

>>> Under a probit link function, a linear predictor of -4 corresponds to a probability of $3e-5$ and a linear predictor of -10 corresponds to a probability of $8e-24$. For comparison, many rare species occur in $1/2874$ (approximately $3e-4$) samples. We note these numbers are not directly comparable with Figure 4, which has normalized other predictors to specific values. What is most relevant for measuring model fit is whether or not our predictions are well calibrated when considering the effects of all features jointly. We have now quantified this by adding a new figure (Fig. 5 in the revision) which compares the expected and observed prevalence across all species, both overall and as a function of key covariates (temperature and precipitation).

In practice, probit and logit links tend to produce nearly identical probabilities. Below we show the two link functions (with the linear predictor of the probit link scaled to make the two curves directly comparable):

(e) There are several other model assumptions that need to be addressed. The covariance matrix of $\text{vec}(\beta)$ and CORAL priors also require justification. Although the covariance matrix of $\text{vec}(\beta)$ and the CORAL priors are convenient and typical model assumptions in a Bayesian framework, I wonder how robust the CORAL model is when the covariance matrix and priors are mis-specified?

>>> This is a very good question, and we have now updated the HMSC and CORAL sections of the paper so they provide more detail on model structure, prior construction, and when key assumptions are likely to hold. We note that CORAL inherits prior structures directly from HMSC, which generally had the highest predictive power compared to other species distribution models (some of which contained alternative prior formulations; Norberg et al. 2019). For this reason – and given the strong out-of-sample performance observed in our work – we expect CORAL to be quite robust to prior misspecification. Ultimately, however, performance will depend on the specific characteristics of the dataset at hand. We have included a new figure (Fig. 5 in the revision) that assesses misspecification by comparing model predictions with data.

(2) Metabarcoding data are often subject to sequencing errors

The authors applied high-throughput COI metabarcoding and the OptimOTU pipeline to record the occurrences of 255,188 OTUs (“species”) in 2874 samples collected in 53 locations across Madagascar. The authors treated those species that occurred in at least 50 samples as “abundant” (there were 876 of them) and the other 254,312 species as “rare”. However, it is well known that sequencing errors often generate spurious low frequency counts, especially singletons, which can lead to an inflated number of rare species and incorrect estimates of diversity (Edgar, 2013; 2017; Bunge et al. 2014). For example, sequencing errors can often cause two sequences that should be classified into the same OTU to be classified as distinct OTUs or as singletons if no match is found. The authors indicated (Line 257 of the manuscript) that there were 182,402 (~ 71% of all species) singletons, which were detected only once in the OTU data. I wonder whether sequencing errors exist in the authors’ data. If a portion of the singletons are spurious, it would not be meaningful to make inferences about species with only one occurrence. To my knowledge, there are at least two statistical methods to “adjust” the number of singletons: Edgar (2018) and Chiu and Chao (2019). These adjusted methods provide estimates of true singletons, which can then be used to assess sample completeness; see Point (3) below.

>>> These concerns are indeed valid with any metabarcoding data. While it is true that in our data, ~ 71% of all species occur in only one sample, it does not mean that they would be represented by singletons (in the sense of being represented by a single sequence copy: this applied to only 1479 species). We thank the Reviewer for pointing out that we had failed to specify this critical information in the manuscript. In response, we have now expanded the section “Case study on Malagasy arthropods” as follows:

“Most of these rare species were extremely rare in the sense that 182,402 species (71% of all rare species) were detected in one sample only. Among these extremely rare species, 1479 were singletons, i.e., represented by a single sequence. Some of these taxa may be artefacts, reflecting chimeric sequences or sequencing error. However, the vast majority (99.4%) of the rare species were represented by more than one sequence. Thus, the potential interpretation of some sequencing errors as false species is unlikely to qualitatively influence our conclusions.”

As the proportion of species represented by singletons was so low (0.006), we decided not to adjust for it, to keep the focus of the manuscript on the core aspects of the CORAL approach.

(3) Assessment of sample completeness

Based on Alan Turing’s statistical perspective, the existence of a large number of singletons implies that many species remain undetected in the data (Good 1953). According to Good (1953), this concept was originally developed by Turing in his cryptographic analysis during World War II. For hyper-diverse arthropod data, it would be useful to quantify sample completeness before any joint species modeling. Conventional sample completeness is defined as the proportion of observed species richness to true richness (observed plus undetected species). However, data are often insufficient to provide an accurate estimate of true species richness. In contrast, Turing’s objective measure of sample completeness is based on the concept of sample coverage (the fraction of an assemblage’s individuals that belong to the observed species). Contrary to most people’s intuition, sample coverage can be accurately estimated based on the sample data itself; Turing’s estimator (one minus the proportion of singletons) is surprisingly simple. For Metabarcoding data, after adjusting for the number of singletons, one can accurately quantify sample completeness. See Slik et al. (2015) for an example.

>>> We thank the Reviewer for this useful suggestion. Nonetheless, since the proportion of singletons was so low, sample coverage would be essentially 100%. For this reason, we have decided not to discuss sample coverage. Instead, we now mention the proportion of species represented as

singletons (see response above). This is indeed critically important information, which we had failed to include in the original submission.

Reviewer #2 (Remarks to the Author):

This paper proposes a new approach (CORAL) for simultaneous inference across multiple species using transfer learning. Important challenges in joint species distribution modelling (JSDM) are handling many species and constructing models for rare species. The proposed approach addresses these by first fitting a "backbone model" to a smaller number of abundant species, then fitting parallel models to rare species using a transfer learning approach, where the model is simplified by taking observed values for latent variables from the backbone model. The method is shown to work very well on a Madagascan arthropod dataset of hundreds of thousands of species, dramatically improving predictive performance. This is an original and useful idea which could have widespread impact in the field. I have no major concerns and comment mostly on the exposition.

>>> We thank the Reviewer very much for these encouraging comments.

CORAL is proposed as an extension of HMSC, a specific software application for fitting JSDMs, but the method is quite general and could be used with other software too. It would have been nice to see some discussion of this, or a formulation of the problem that was not HMSC-specific.

>>> We have now added the following description to justify why we present CORAL specifically in the HMSC context: "The core idea of CORAL is very general and will thus apply to many other JSDM approaches, too. What makes its application in the HMSC context so intuitive is that HMSC models species responses to predictors as a function of species traits and phylogenetic relationships. This feature can be efficiently harnessed for transfer learning."

Some guidance for users would be helpful - is there a software implementation of this approach? If not, is there some simple example code you could share for how to implement this approach?

>>> Below, the reviewer writes that "I partly reviewed the code - I looked at the demo simulated dataset". For this reason, we are a bit unsure what the reviewer means, as the demonstration was indeed intended to provide a simple example code for how to implement this approach. To make it more apparent how our approach can be implemented, we have now revised the software implementation and its description. Specifically, we have integrated the CORAL approach to the R-package Hmsc and included three software demonstrations in the Zenodo repository. These demonstrate how to use the software with a small, simulated dataset, with running time less than a minute in a typical laptop (Demo 1); (2) how to apply the analyses presented in the paper for a small subset of the data, with running time of ca. one hour in a powerful laptop (Demo 2); how to reproduce the full analyses presented in the paper, with running time up to several days, depending on the computational resources (Demo 3). The Demos 1 and 2 are aimed to be user-friendly starting points for understanding and testing how to implement CORAL. The Demo 3 is included mainly for reproducibility. We have also added a graphical description of the full statistical approach in Fig. 1, (panel B), which we hope will help the readers to connect the software implementation to the statistical approach.

The Methods section only describes the computational methods used in very general terms, while going into a lot of detail about the experimentation and bioinformatics pipeline used to collect and pre-process the data. More information about the statistical procedures used is needed, especially given that this is the main focus of the paper.

>>> We agree that the original submission included a lot of detail about experimentation, DNA extraction and sequencing, even though these aspects are not at the core of the manuscript. During the review process, we published two protocol papers (one about the empirical sampling protocol, and one about the sequencing protocol), and one about the bioinformatics pipeline. Thus, we are now able to refer to these three papers. This has enabled us to substantially shorten the corresponding parts of the methods, thus making the manuscript more focused on the CORAL method itself. To provide more detail on the statistical procedures, we have now updated the HMSC section and significantly revised the CORAL section. This revision includes the augmentation of Figure 1 with the complete statistical model structure.

The idea of transfer learning is key to the paper but it is not explained to readers at all, only a reference is given to a paper on Bayesian transfer learning. It would be helpful to provide some context here, and explanation of how transfer learning works, when it was proposed, where it is typically used.

>>> We thank the Reviewer for this important suggestion. In response, we have added additional context to the start of the Results, where CORAL is formally introduced.

It is stated a couple of times (lines 83, 285) that because the proposed approach is scalable to millions of species it is capable of modelling all of Earth's biodiversity... this is an attractive idea but I feel like there is some overreach here. This approach only deals with a large number of responses (species) not a large number of observations (sites/replicates). To model all of Earth's biodiversity we would need a pretty large sample size, in order to capture the diversity of environments in which we find species, and there would be further technical challenges to overcome in doing this (including capturing spatial structure, which would vary at a relatively fine scale in a global analysis). Perhaps some minor rephrasing could clear this up.

>>>We agree and have rephrased the corresponding section.

Reviewer #2 (Remarks on code availability):

I partly reviewed the code - I looked at the demo simulated dataset - and was confused as to why the second stage backbone model fit (which I believe is predicting rare species, using the backbone model constructed from abundant species?) happened before calculation of the prior variance for rare species? Don't you need the prior variance in order to fit the rare species model? Perhaps the workflow for implementing CORAL could be clarified...

>>> As mentioned above, we have made the software much more user friendly by integrating the CORAL approach into the R-package Hmsc, which has simplified the provided example scripts and their documentation. We are confident that these additions resolve the questions raised by the Reviewer.

Reviewer #3 (Remarks to the Author):

This paper develops and tests a statistical framework to predict the distribution of species based on environmental predictors. One problem for this kind of species distribution models is that rare species often do not have enough data to allow robust prediction. This paper addresses the issue by using phylogenetic relationships between species to infer the environmental parameters of the rare species, though a bayesian statistical framework. I think that the approach is novel and has an interesting potential. But I have some major concerns about the current draft of the paper as I detail below.

>>> We thank the reviewer for these encouraging comments and for providing critical comments which have helped us improve our work.

General Comments

1. The statistical approach used by the model is arguably very complex, and I had difficulty in following it at several times. I found that some of the explanations could be made more understandable. In addition it is not clear that the complexity is always warranted. For instance, the authors attempt to make the approach as generic as possible to also allow for the use of traits. But then they do not use the trait matrix in the application of the model, so not only we don't know if it can be effective, it adds a layer of complexity that is not ultimately needed in this paper.

>>> While it is true that our specific case study does not utilize traits, we have implemented the CORAL approach (and provided software) for the full HMSC model that includes traits and phylogenetic relationships. As some users will be keen to utilize trait data, we believe that it is important to include this feature. To clarify the relation between the case examples raised in our paper and the full scope of the CORAL approach, we have significantly expanded the CORAL section (including Figure 1). We hope and trust that this has improved the clarity of our goals and approach.

2. A second issue related to the complexity of the approach is the issue of reusability of this approach. We are provided with some R code (which I did not have time to examine), but it seems to imply access to massive computational power. I would like to have seen a small version of the model/test dataset for the less resourced scientists to examine.

>>> To clarify this issue, let us first note that in the original submission, we had provided two demonstration components: a small software demonstration that runs on a typical laptop in a few seconds, and a reproducible example of the empirical case study. The latter does indeed require substantial computational power. To improve on this aspect, we have now made the software much more user friendly by integrating the CORAL approach to the R-package Hmsc. As detailed above and in the Zenodo repository, we now provide three software demonstrations.

3. Test of the approach. I found it challenging to assess the testing of the approach. I am not sure I understand what is shown in Figures 3 and 4 and the range of metrics chosen. In addition I would like to have seen a test of the model with data for which we had a better way of testing the model fit. For instance, one could have a subset of common species for which a subsample would be generated. We know for those species what are the predicted occurrences and what are the important predictor variables. Is this inference approach based on phylogenetics good at picking those predictor variables? Alternatively a synthetic dataset would also allow us to assess the model fit for known/controlled conditions.

>>> To clarify what is being shown, we have now expanded the captions for Figures 3 and 4. We also point the reader to the methods for full details regarding performance metrics and their interpretations. To clarify why these metrics provide a detailed summary of classifier performance, we have expanded the methods. We now also include a new figure of posterior predictive checks comparing the estimated and observed prevalences, both overall and as a function of key covariates.

Unfortunately, we are not entirely sure why the Reviewer considers our test of the model less than ideal. In our view, the test that presented is indeed the ideal one, because we specifically test the ability of the CORAL approach (common to rare transfer learning) to predict the occurrences of rare species based on information from common species. Our results show that this approach predicts the occurrence of the rare species much better than by random – even without seeing any data on

the rare species. This we consider to be a highly non-trivial demonstration of the power of the approach. We further note that in the original submission, we included a software demonstration with simulated data. This demonstration showed the validity of the approach for known conditions.

Specific Comments

I.38. We don't know how many species there are in the planet. Most estimates suggest between 4 and 10 Million, but given a single number is misleading.

>>> We agree and have rephrased.

I.111. The use of T to represent the matrix of traits and the transpose operation is very confusing. Suggest use " t " for transpose or change the letter for the trait matrix.

We agree; however, both are established notations within HMSC and in the broader mathematical literature, so we believe it would be confusing to change them. We now clarify that the superscript T denotes the matrix transpose.

I.106, 111. Why are B and μ vectorised? Is this for computational purpose or analytical purposes?

>>> This is for analytical purposes, as described in more detail in the references provided.

I.183-184. The choice of the climate predictors seems a bit limited (annual values only) and why was a semestral seasonality added to the annual seasonality?

>>> We agree that we only use a limited set of climatic predictors. This is because we wanted to keep the model simple, and because we included site-level random effects that are able to account for missing predictors related to climatic or other conditions which vary among the sites. The semestral seasonality was added to avoid the assumption that the time of the year that leads to highest probability of detection is exactly half-a-year apart from the time of the year that leads to lowest probability of detection. Such an assumption would not be biologically justified.

Figure 2. Why are there only 4 samples and 10 sites? Is this a subset of the study?

>>> The model does not involve 4 samples but 4 latent factors at the sample level. In a similar vein, it does not involve 10 sites, but 10 latent factors at the site level. The model indeed includes all samples from all sites. Lead by the Reviewer's comment, we realized that our original labelling of the figure was confusing. This feature has now been corrected.

Manuscript ID: NMETH-A56224

Title: Common to rare transfer learning (CORAL) enables inference and prediction for a quarter million rare Malagasy arthropods

Journal: *Nature Methods*

Dear Senior Editor Nina Vogt, PhD

Thank you very much for the careful assessment of our manuscript and for inviting us to submit a minor version. We have now responded to all remaining reviewer comments. Below, we provide a point-by-point response to all the comments. The original comments are shown in *italics* and our responses in regular font and starting with ">>>".

We trust you will find the current version of the manuscript publishable in *Nature Methods*.

We wish to participate in transparent peer review.

Kind regards,

Prof. Otso Ovaskainen and co-authors.

Reviewer #1 (Remarks to the Author):

In the revision, the authors have properly responded or implemented my questions/comments raised in my earlier review. I have no further comments.

>>> We are happy to hear that the reviewer is satisfied with our responses.

Reviewer #2 (Remarks to the Author):

I have reviewed the author response, the changes to the manuscript in response to reviewer comments, and some of the new demo code demonstrating how to apply CORAL in practice. I think the authors have done an excellent job responding to reviewer comments and have no further suggestions.

>>> We are happy to hear that the reviewer is satisfied with our responses.

Reviewer #3 (Remarks to the Author):

I thank the authors for addressing some of my comments in a previous draft. However I don't think they addressed fully one of my major comments regarding the testing of the approach. The authors just argue they are using the ideal approach to test the data. I don't think that predicting the occurrence of rare species better than by random is the "ideal" approach (random seems like a low bar to me, but maybe I am missing something). My proposal was to use a subsetting of the data for the test, a bit like in a training/validation split, but using "rarified" occurrences from the common species data.

>>> We thank the reviewer for re-explaining their suggestion. It appears that we did not fully understand the reviewer's suggestion at the first time, and we apologize for that. We have now implemented the analyses specifically proposed by the reviewer. Namely, we have applied the CORAL approach to rarefied occurrences of the common species and compared the estimated

environmental responses to those identified by the full HMSC model. We report the results of this analysis in the following section that we have added to the Results: “To further validate the ability of the CORAL approach to infer the environmental responses of the rare species, we rarefied the occurrence data on the common species by masking 90% of their occurrences, thus simulating the case that the common species would actually be rare species. We then used the CORAL approach to estimate the environmental responses of the common species from the rarefied data and compared these estimates to their environmental responses estimated by the backbone model fitted to the full data. The CORAL approach was generally successful in estimating the environmental responses of the species, as there was a high correspondence between the two kinds of estimates: the mean (standard deviation) correlation was 0.68 (0.06) for climatic and seasonal predictors, 0.69 (0.04) for sample-level latent factors, and 0.73 (0.04) for site-level latent factors.”

They say they provided "a software demonstration with simulated data" that showed "the validity of the approach for known conditions" but fail to point out exactly in what page of the manuscript this is done. So I would still like to see this issue better addressed or explained.

>>> The software demonstration with simulated data is part of the Zenodo repository (10.5281/zenodo.11076832) that we link to in the manuscript. In the previous version of the manuscript, we failed to mention that the repository includes a simulated data case study that shows the validity of the approach for known conditions. We now explicitly mention this: “To enable easy application of the CORAL approach to high-dimensional biodiversity data, we provide an R package for fitting these models, visualizing the results, and generating predictions (see Code availability). This software package also includes a simulated case study that demonstrates how the CORAL approach is able to recover the true parameter values that were used to simulate the data.”